# DeepLag: Discovering Deep Lagrangian Dynamics for Intuitive Fluid Prediction

**Qilong Ma,**[*] **Haixu Wu,**[*] **Lanxiang Xing, Shangchen Miao, Mingsheng Long**[✉]

School of Software, BNRist, Tsinghua University, China

{mql22,wuhx23,xlx22,msc21}@mails.tsinghua.edu.cn, mingsheng@tsinghua.edu.cn

## Abstract

Accurately predicting the future fluid is vital to extensive areas such as meteorology, oceanology, and aerodynamics. However, since the fluid is usually observed from the Eulerian perspective, its moving and intricate dynamics are seriously obscured and confounded in static grids, bringing thorny challenges to the prediction. This paper introduces a new Lagrangian-Eulerian combined paradigm to tackle the tanglesome fluid dynamics. Instead of solely predicting the future based on Eulerian observations, we propose DeepLag to discover hidden Lagrangian dynamics within the fluid by tracking the movements of adaptively sampled key particles. Further, DeepLag presents a new paradigm for fluid prediction, where the Lagrangian movement of the tracked particles is inferred from Eulerian observations, and their accumulated Lagrangian dynamics information is incorporated into global Eulerian evolving features to guide future prediction respectively. Tracking key particles not only provides a transparent and interpretable clue for fluid dynamics but also makes our model free from modeling complex correlations among massive grids for better efficiency. Experimentally, DeepLag excels in three challenging fluid prediction tasks covering 2D and 3D, simulated and real-world fluids. Code is available at this repository: https://github.com/thuml/DeepLag.

## 1 Introduction

Fluids, characterized by a molecular structure that offers no resistance to external shear forces, easily deform even under minimal stress, leading to highly complex and often chaotic dynamics [10]. Consequently, the solvability of fundamental theorems in fluid mechanics, such as Navier-Stokes equations, is constrained to only a limited subset of flows due to their inherent complexity and intricate multiphysics interactions [38]. In practical applications, computational fluid dynamics (CFD) is widely employed to predict fluid behavior through numerical simulations, but it is hindered by significant computational costs. Accurately forecasting future fluid dynamics remains a formidable challenge. Recently, deep learning models [8, 28, 22] have shown great promise for fluid prediction due to their exceptional non-linear modeling capabilities. These models, trained on CFD simulations or real-world data, can serve as efficient surrogate models, dramatically accelerating inference.

A booming direction for deep fluid prediction is learning deep models to solve partial differential equations (PDEs) [45]. However, most of these methods [48, 29, 22, 21] attempt to capture fluid dynamics from the Eulerian perspective, which means modeling spatiotemporal correlations among massive grids unchanging over time. From this perspective, the complicated moving dynamics in fluids could be seriously obscured and confounded in static grids, bringing challenges in both computational efficiency and learning difficulties for accurately predicting future fluids.

In parallel to the Eulerian method, we notice another major approach for elucidating fluid dynamics, the Lagrangian method [13], also known as the particle tracking method. This method primarily

---

[*]Equal Contribution

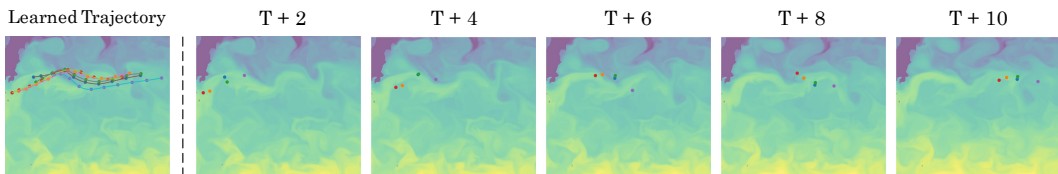

| Learned Trajectory | T + 2 | T + 4 | T + 6 | T + 8 | T + 10 |

Figure 1: Comparison between Lagrangian (left) and Eulerian (right) perspectives. The left depicts the learned trajectories of Lagrangian particles overlaid on the mean state, while the right displays the positions of tracked particles in successive Eulerian frames. Fluid motion is more visibly represented through the dynamic Lagrangian view compared to the density variations in static Eulerian grids.

focuses on tracing individual fluid particles by modeling the temporal evolution of their position and velocity. Unlike the Eulerian methods, which observe fluid flow at fixed spatial locations, the Lagrangian approach describes the fluid dynamics through the moving trajectory of individual fluid particles, offering a more natural and neat representation of fluid dynamics with inherent advantages in capturing intricate flow dynamics. Moreover, it allows a larger inference time step tha Eulerian methods while complying with the Courant–Friedrichs–Lewy condition [6] that guarantees stability. In Figure 1, we can find that fluid dynamics is much more visually apparent in Lagrangian trajectories on the left compared to the density changes observed on static Eulerian grids on the right.

Building on the two perspectives mentioned earlier, we propose DeepLag as a Eulerian-Lagrangian Recurrent Network. Our aim is to integrate Lagrangian tracking into the deep model, thereby enhancing the dynamics modeling in Eulerian fluid prediction. To achieve this, we present the EuLag Block, a powerful module that accomplishes Lagrangian tracking and Eulerian predicting at various scales. By leveraging the cross-attention mechanism, the EuLag Block assimilates tracked Lagrangian particle dynamics into the Eulerian field, guiding fluid prediction. It also forecasts the trajectory and dynamics of Lagrangian particles with the aid of Eulerian features. This unique Eulerian-Lagrangian design harnesses the dynamics information captured by Lagrangian trajectories and the fluid-structure features learned in the Eulerian grid. In our experiments, DeepLag consistently outperforms existing models, demonstrating state-of-the-art performance across three representative datasets, covering 2D and 3D fluid at various scales. Our contributions are as follows:

- Going beyond learning fluid dynamics at static grids, we propose DeepLag featuring the Eulerian-Lagrangian Recurrent Network, which integrates both Eulerian and Lagrangian frameworks from fluid dynamics within a pure deep learning framework concisely.

- Inspired by Lagrangian mechanics, we present EuLag Block, which can accurately track particle movements and interactively utilize the Eulerian features and dynamic information in fluid prediction, enabling a better dynamics modeling paradigm.

- DeepLag achieves consistent state-of-the art on three representative fluid prediction datasets with superior trade-offs for performance and efficiency, exhibiting favorable practicability.

## 2 Preliminaries

### 2.1 Eulerian and Lagrangian Methods

Eulerian and Lagrangian descriptions are two fundamental perspectives for modeling fluid motion. The Eulerian view, commonly used in practical applications [23], observes fluid at fixed points and records physical quantities, such as density, as a function of position and time, $\mathbf{v} = \mathbf{v}(\mathbf{s}, t)$. Thus, future fluid can be predicted by integrating velocity along the temporal dimension and interpolating the results to observed grid points [49]. In contrast, the Lagrangian view focuses on the trajectory of individual particles, tracking a particular particle from initial position $\mathbf{s}_0$ by its displacement $\mathbf{d} = \mathbf{d}(\mathbf{s}_0, t)$ at time $t$. This approach reveals the intricate evolution of the fluid by following particle trajectories, making it convenient for describing complex phenomena like vortices, turbulence, and interface motions [40]. Two perspectives are constitutionally equivalent, as bridged by the velocity:

$$\mathbf{v}\left(\mathbf{d}(\mathbf{s}_0, t), t\right) = \frac{\partial \mathbf{d}}{\partial t}\left(\mathbf{s}_0, t\right). \tag{1}$$

Furthermore, the *material derivative* $\frac{\mathrm{D}\mathbf{q}}{\mathrm{D}t}$ that describes the change rate of a physical quantity $\mathbf{q}$ of a fluid parcel can be written as the sum of the two terms reflecting the spatial and temporal influence on $\mathbf{q}$ [1], which represent the derivatives on Eulerian domain and Lagrangian convective respectively:

$$\frac{\mathrm{D}\mathbf{q}}{\mathrm{D}t} \equiv \underbrace{\frac{\partial \mathbf{q}}{\partial t}}_{\text{Domain derivative}} + \underbrace{\mathbf{u} \cdot \nabla \mathbf{q}}_{\text{Convective derivative}} . \tag{2}$$

This connection inspires us to incorporate Lagrangian descriptions into dynamics learning with Eulerian data, enabling a more straightforward decomposition of complex spatiotemporal dependencies.

While traditional particle-based (or mixed-representation) solvers demonstrate superior accuracy and adaptability in inferring small-scale phenomena and dealing with nonlinear and irregular boundary conditions, they require computing the acceleration of each particle through physical equations, followed by sequential updates of their velocity and position [31]. This pointwise modeling approach often demands a significant number of points to fully characterize the dynamics of the entire field to meet accuracy requirements. Moreover, the irregular arrangement of particles and difficulty in parallelization result in higher computational costs and challenges with particle interpolation and gridding [12]. This renders particle-based solvers suboptimal compared to Eulerian solvers, particularly in high-dimensional spaces and large-scale simulations. However, the proposed DeepLag leverages the strengths of both solvers, eschewing equations and instead utilizing Eulerian information to assist particle tracking directly. This greatly alleviates the pressure on Lagrangian representation, requiring significantly fewer representative particles to aggregate the dynamics of the entire field.

## 2.2 Neural Fluid Prediction

As computational fluid dynamics (CFD) methods often require hours or even days for simulations [41], deep models have been explored as efficient surrogate models that can provide near-instantaneous predictions. These neural fluid prediction models approximate the solutions of governing fluid equations differently and can be categorized into three mainstream paradigms as in Figure 2(a-c).

**Classical ML methods**   As depicted in Figure 2(a), these methods either replace part of a numerical method with a neural surrogate [39] or encode multivariate fields into a single-variable latent state $\mathbf{z}$ [4, 26, 55], on which they model an ODE governing the state function $\mathbf{z}_t : \mathcal{T} \rightarrow \mathbb{R}^d$, representing the first-order time derivative, through neural networks. However, the absence of physical meaning and guidance for latent states in evolution leads to error accumulation and a generally short forecasting horizon. A detailed comparison between DeepLag and these models is provided in Appendix B.

**Physics-Informed Neural Networks (PINNs)**   This branch of methods in Figure 2(b) adopts deep models to learn the mapping from coordinates to solutions and formalize PDE constraints along with initial and boundary conditions as the loss functions [48, 29, 46, 47]. Though this paradigm can explicitly approximate the PDE solution, they usually require exact formalization for coefficients and conditions, limiting their generality and applicability to real-world fluids that are usually partially observed [35]. Plus, the Eulerian input disables them from handling Lagrangian descriptions.

**Neural Operators**   Recently, a new paradigm, illustrated in Figure 2(c), has emerged where deep models learn the neural operators between input and target functions, e.g., past observations to future fluid predictions. Since DeepONet [22], various neural operators have significantly advanced fluid prediction, which directly approximates mappings between equation parameters and solutions. For Eulerian grid data, models based on U-Net [32] and ResNet [15] architectures have been proposed [30, 27, 17], as well as variants addressing issues like generalizing to unseen domains [44], irregular mesh [11], and uncertainty quantification [51]. Transformer-based models [43] enhance modeling capabilities and efficiency by exploiting techniques like Galerkin attention [3], incorporating ensemble information from the grid [14], applying low-rank decomposition to the attention mechanism [20], and leveraging spectral methods in the latent space [52]. Additionally, FNO [21] learns mappings in the frequency domain, and MP-PDE [2] utilizes the message-passing mechanism. For Lagrangian fluid particle data, some CNN-based methods [34, 42] model particle interactions through redesigned basic modules, while GNN-based methods [33, 25] update particle positions using the Encode-Process-Decode paradigm. Despite the progress made by these methods, they are limited to one description and do not seamlessly combine Eulerian and Lagrangian views.

# 3 DeepLag

Following the convention of neural fluid prediction [21], we formalize the fluid prediction problem as learning future fluid given past observation, as shown in Figure 2(d). Given a bounded open subset of $d$-dimensional Euclidean space $\mathcal{D} \subset \mathbb{R}^d$ and a Eulerian space $\mathcal{U} \subset \mathbb{R}^o$ with $o$ observed physical quantities, letting $\mathbf{u}_t(\mathbf{x}) \subset \mathcal{U}$ and $\mathbf{u}_{t+1}(\mathbf{x}) \subset \mathcal{U}$ represent the Eulerian fluid field observation at two consecutive time steps on a finite coordinate set $\mathbf{x} \subset \mathcal{D}$, we aim at fitting the mapping $\Phi : \mathbf{u}_t(\mathbf{x}) \to \mathbf{u}_{t+1}(\mathbf{x})$. Concretely, provided initial $P$ step observations $U_P = \{\mathbf{u}_1, \ldots, \mathbf{u}_P\}$, the fluid prediction process can be written as the following autoregressive paradigm:

$$U_t = \{\mathbf{u}_{t-P+1}, \ldots, \mathbf{u}_t\} \xrightarrow{\mathcal{F}_\theta} \mathbf{u}_{t+1}, \tag{3}$$

where $t \geq P$ and $\mathcal{F}_\theta$ represents the learned mapping between $U_t$ and the predicted field $\mathbf{u}_{t+1}$.

Inspired by the material derivative in Eq. (2), we present DeepLag as a Eulerian-Lagrangian Recurrent Network, which utilizes the EuLag Block to learn Eulerian features and Lagrangian dynamics interactively at various scales to address the complex spatiotemporal correlations in fluid prediction. Specifically, we capture the temporal evolving features at fixed points from the Eulerian perspective and the spatial dynamic information of essential particles from the Lagrangian perspective through their movement. By integrating Lagrangian pivotal dynamic information into Eulerian features, we fully model the spatiotemporal evolution of the fluid field over time and motion. Moreover, DeepLag can obtain critical trajectories within fluid dynamics with high computational efficiency by incorporating high-dimensional Eulerian space into lower-dimensional Lagrangian space.

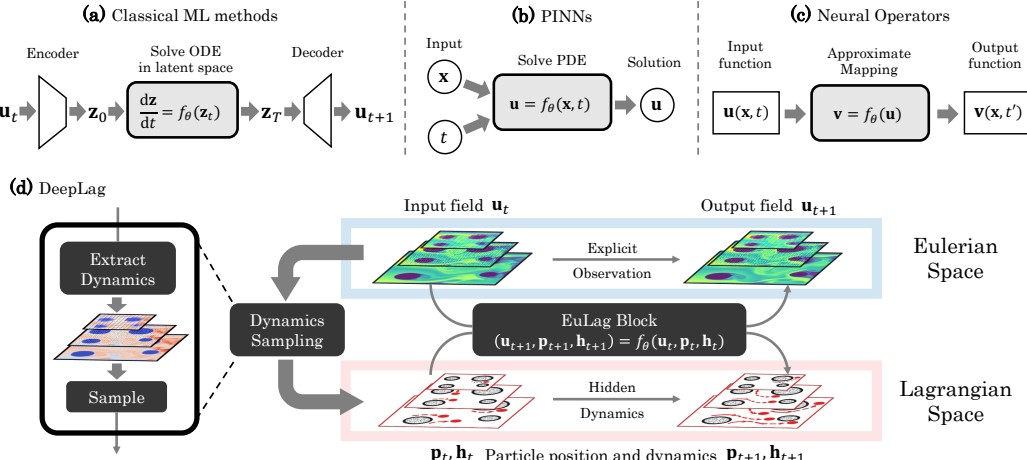

Figure 2: Three types of neural fluid prediction models (a-c) and overview of DeepLag (d). The EuLag Block accumulates the previous dynamics at each time and scale to guide the Eulerian field update and then evolves the particle movement and dynamics conditioned on the updated field.

## 3.1 Overall Framework

It is widely acknowledged that fluids exhibit different motion characteristics across varying scales [3, 52]. In order to capture the intrinsic dynamics information at different scales, we track the trajectories of the key particles on $L$ scales separately and propose a Eulerian-Lagrangian Recurrent Network to realize the interaction between Eulerian and Lagrangian information. For clarity, we omit the scale index $l$ for primary physical quantities in this subsection, where $l \in \{1, 2, \cdots, L\}$.

**Initializing Lagrangian particles** To better capture complex dynamics in the fluid field, we sample by importance of the points from Eulerian perspective observations to determine initial positions for Lagrangian tracking. This is done by our dynamics sampling module. For the first predicting step $t = P$, given observations $\mathbf{u}_t = \{\mathbf{u}_t(\mathbf{x}_k) | \mathbf{x}_k \in \mathcal{D}_l, 1 \leq k \leq N_l\} \in \mathbb{R}^{N_l \times o}$ on all $N_l$ points in observation domain $\mathcal{D}_l \subset \mathbb{R}^d$ of each scale, we extract their spatial dynamics by a convolutional network. We then calculate the probability matrix $\mathbf{S} \in \mathbb{R}^{N_l}$ via softmax along spatial dimension:

$$\mathbf{S} = \text{Softmax}\left(\text{ConvNet}(\mathbf{u}_t)\right), \tag{4}$$

where $\text{ConvNet}(\cdot)$ consists of a convolutional layer and a linear layer, with activation function in-between. We then sample $M_l$ Lagrangian tracking particles by the probability matrix at each scale:

$$\mathbf{p}_t = \text{Sample}(\{\mathbf{x}_k\}_{k=1}^{N_l}, \mathbf{S}), \tag{5}$$

where $\mathbf{p}_t = \{\mathbf{p}_{t,i}^l \in \mathcal{D}_l\}_{i=1}^{M_l} \in \mathbb{R}^{M_l \times d}$ represents the set of sampled $M_l$ particles.

**Eulerian fluid prediction with Lagrangian dynamics** For the $t$-th timestep, we adopt a learnable embedding layer with multiscale downsampling $\text{Down}(\cdot)$ to encode past observations or predictions $U_t = \{\mathbf{u}_{t-P+1}, \ldots, \mathbf{u}_t\}$ to obtain the Eulerian representations $\mathbf{u}_t^l \in \mathbb{R}^{N_l \times C_l}$ at each scale.

We integrate the EuLag Block to fuse the Lagrangian dynamics at each scale and direct the evolution of Eulerian features. This interaction simultaneously enables Eulerian features to guide the progression of Lagrangian dynamics. For the $l$-th scale, we track $M_l$ key particles over time, with $\mathbf{p}_t = \{\mathbf{p}_{t,i}^l \in \mathcal{D}_l\}_{i=1}^{M_l}$ representing their positions and $\mathbf{h}_t = \{\mathbf{h}_{t,i}^l \in \mathbb{R}^{C_l}\}_{i=1}^{M_l}$ denoting their learned particle dynamics. As shown in Figure 2(d), the positions and dynamics of the $M_l$ particles at the $l$-th scale are learned autoregressively using the EuLag Block, which can be written as

$$\mathbf{u}_{t+1}, \{\mathbf{p}_{t+1}\}, \{\mathbf{h}_{t+1}\} = \text{EuLag}(\mathbf{u}_t, \{\mathbf{p}_t\}, \{\mathbf{h}_t\}), \tag{6}$$

where the scale index $l$ is omitted for notation simplicity. The EuLag Block learns to optimally leverage the complementary strengths of Eulerian and Lagrangian representations, facilitating mutual refinement between these two perspectives towards an exceeding performance. More details about the specific implementation of the EuLag Block are elaborated in the following subsection 3.2.

After evolving into new Eulerian features $\mathbf{u}_{t+1}$, Lagrangian particle position $\mathbf{p}_{t+1}$ and dynamics $\mathbf{h}_{t+1}$ at the $l$-th scale, we further aggregate $\mathbf{u}_{t+1}$ with the predicted Eulerian field at a coarser scale by upsampling $\text{Up}(\cdot)$. Eventually, the full-resolution prediction $\mathbf{u}_{t+1}$ at step $t+1$ is decoded from $\mathbf{u}_{t+1}^1$ with a projection layer. We unfold the implementation of the overall architecture in Appendix A.2.

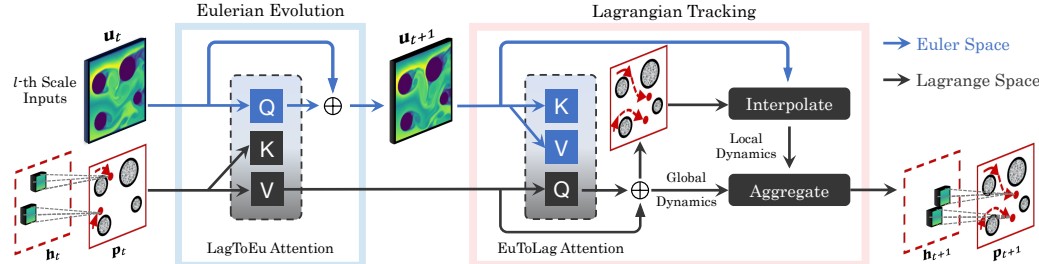

Figure 3: Overview of the EuLag Block, which accumulates previous dynamics information to guide Eulerian evolution for predicting particle movement. Scale index $l$ is omitted for simplicity.

## 3.2 EuLag Block

As stated in Eq. (6), we adopt a recurrent network to interactively exploit the information from two fluid descriptions, which consist of two main components: Lagrangian-guided feature evolving and Eulerian-conditioned particle tracking. The following quantities are all in the $l$-th scale.

**Lagrangian-guided feature evolving** Classical theories [9] and numerical algorithms [31] show that fluid predictions can be solved by identifying the origin of the fluid parcel and interpolating the dependent variable from nearby grid points. However, without specifying certain PDEs, we cannot explicitly determine the former position of the particle on each Eulerian observed point. Thus, we first adaptively synthesize the Lagrangian dynamics of the tracked particles to guide the evolution of Eulerian features using a cross-attention mechanism. Formally, we adopt a Lagrangian-to-Eulerian cross-attention, where the Eulerian field $\mathbf{u}_t$ serves as queries, and Lagrangian dynamics concatenated with particle position $\mathbf{h}_t||\mathbf{p}_t \in \mathbb{R}^{M_l \times (C_l + d)}$ is used as keys and values:

$$\text{LagToEu-Attn}(\mathbf{u}_t, \mathbf{h}_t||\mathbf{p}_t, \mathbf{h}_t||\mathbf{p}_t) = \text{Softmax}\left(\frac{\mathbf{W}_Q \mathbf{u}_t (\mathbf{W}_K \cdot \mathbf{h}_t||\mathbf{p}_t)^\top}{\sqrt{C_l + d}}\right) \mathbf{W}_V \cdot \mathbf{h}_t||\mathbf{p}_t, \tag{7}$$

where $\mathbf{W}_Q$, $\mathbf{W}_K$ and $\mathbf{W}_V$ stand for linear projections. We wrap the attention in a Transformer block with residual connections to implement the Lagrangian-dynamics-guided Eulerian evolution process:

$$\mathbf{u}_{t+1} = \mathbf{u}_t + \text{LagToEu}(\mathbf{u}_t, \mathbf{h}_t||\mathbf{p}_t, \mathbf{h}_t||\mathbf{p}_t). \tag{8}$$

**Eulerian-conditioned particle tracking**   Traditional Lagrangian methods rely on interactions among vast quantities of particles to estimate the future fluid fields. However, the high computational cost hinders the application in deep surrogate models. In other data-driven approaches, the sparse sampling of particles is computational-friendly but cannot directly derive Lagrangian dynamics for other particles. Considering the equivalence of the Eulerian and Lagrangian representations indicated by the material derivative in Eq. (1), we propose to learn particle movements based on the Eulerian conditions. Concretely, we utilize another Eulerian-to-Lagrangian cross-attention, where the evolved dense Eulerian features are used to navigate the Lagrangian dynamics of sparse particles:

$$\text{EuToLag-Attn}(\mathbf{h}_t||\mathbf{p}_t, \mathbf{u}_{t+1}, \mathbf{u}_{t+1}) = \text{Softmax}\left(\frac{(\mathbf{W}'_Q \cdot \mathbf{h}_t||\mathbf{p}_t)(\mathbf{W}'_K \mathbf{u}_{t+1})^\mathsf{T}}{\sqrt{C_l}}\right)\mathbf{W}'_V \mathbf{u}_{t+1}, \quad (9)$$

where we use a different set of $\mathbf{W}'_Q$, $\mathbf{W}'_K$, and $\mathbf{W}'_V$. Similarly, the Transformer block EuToLag wrapping this attention produces the change of forecasted global Lagrangian dynamics $\delta\mathbf{h}_{\text{global},t}$ and movement of tracking particles $\delta\mathbf{p}_t$, which leads to the next step by residual connections:

$$\mathbf{h}_{\text{global},t+1}||\mathbf{p}_{t+1} = \mathbf{h}_t||\mathbf{p}_t + \text{EuToLag}(\mathbf{h}_t||\mathbf{p}_t, \mathbf{u}_{t+1}, \mathbf{u}_{t+1}). \quad (10)$$

To better model the dynamic evolution of the particles, we gather local Lagrangian dynamic by employing bilinear interpolation to evolved Eulerian features $\mathbf{u}_{t+1}$ on new particle position $\mathbf{p}_{t+1}$, then use a linear function to aggregate it with global dynamics information $\mathbf{h}_{\text{global},t+1}$:

$$\mathbf{h}_{t+1} = \text{Aggregate}(\text{Interpolate}(\mathbf{u}_{t+1}, \mathbf{p}_{t+1}), \mathbf{h}_{\text{global},t+1}). \quad (11)$$

Additionally, particles could move out of the observation domain as the input field may have an open boundary. We check the updated position of tracking particles and resample from the latest probability matrix $\mathbf{S}$ to substitute the ones that exit, ensuring the validity of the Lagrangian information.

Overall, the EuLag Block can fully utilize the complementary advantages of Eulerian and Lagrangian perspectives in describing fluid dynamics, thereby being better suited for fluid prediction. For more implementation details of the EuLag Block, please refer to Appendix A.3.

## 4   Experiments

We evaluated DeepLag on three challenging benchmarks, including simulated and real-world scenarios, covering both 2D and 3D, as well as single and multi-physics fluids. Following the previous convention [21], we train DeepLag and the baselines for each task to predict ten future timesteps in an autoregressive fashion given ten past observations. Detailed benchmark information is listed in Table 1. We provide an elaborate analysis of the efficiency, parameter count, and performance difference in section 4.4 and Appendix E. Additionally, more detailed visualizations, besides in later this section, are provided in Appendix H. Furthermore, the trained models are engaged to perform 100 frames extrapolation to examine their long-term stability, whose results are in Appendix I.

**Baselines**   To demonstrate the effectiveness of our model, we compare DeepLag with seven baselines on all benchmarks, including the classical multiscale model U-Net [32] and advanced neural operators for Navier-Stokes equations: FNO [21], Galerkin Transformer [3], Vortex for 2D image [7], GNOT [14], LSM [52] and Fact-Former [20]. U-Net has been widely used in fluid modeling, which can model the multiscale property precisely. LSM [52] and FactFormer [20] are previous state-of-the-art neural operators.

Table 1: Summary of the benchmarks. #Var refers to the number of observed physics quantities in fluid. #Space is the spatial resolution.

| Datasets | Type | #Var | #Dim | #Space |
|---|---|---|---|---|
| Bounded N-S | Simulation | 1 | 2D | $128 \times 128$ |
| Ocean Current | Real World | 5 | 2D | $180 \times 300$ |
| 3D Smoke | Simulation | 4 | 3D | $32^3$ |

**Metrics**   For all three tasks, we follow the convention in neural fluid prediction [21, 52] and report relative L2 as the main metric. Implementations of the metrics are included in Appendix A.4.

**Implementations**   Aligned with convention and the baselines, DeepLag is trained with relative L2 as the loss function on all benchmarks. We use the Adam [18] optimizer with an initial learning rate of $5 \times 10^{-4}$ and StepLR learning rate scheduler. The batch size is set to 5, and the training process is stopped after 100 epochs. All experiments are implemented in PyTorch [24] and conducted on a single NVIDIA A100 GPU. Training curves are shown in Appendix D.

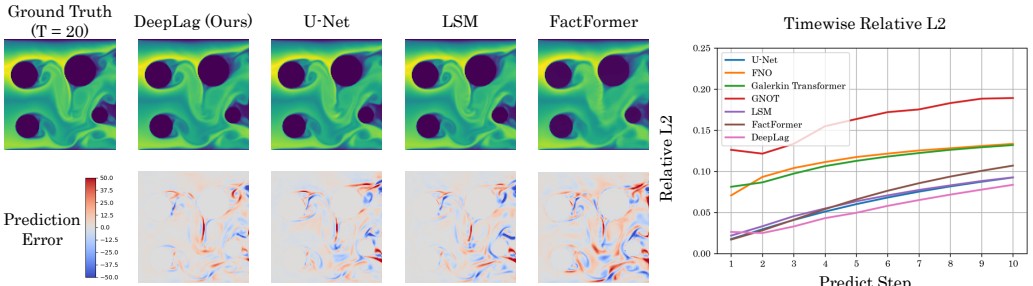

Figure 4: Showcases (left) and timewise relative L2 (right) on Bounded Navier-Stokes dataset. Both predictions (upper row) and absolute error maps (lower row) are plotted for intuitive comparison.

## 4.1 Bounded Navier-Stokes

**Setups**   In real-world applications, handling complex boundary conditions in predicting fluid dynamics is indispensable. Thus, we experiment with the newly generated Bounded Navier-Stokes, which simulates a scenario where some colored dye flows from left to right through a 2D pipe with several fixed pillars as obstacles inside. Details about this benchmark can be found in Appendix C.1.

**Quantitive results**   As shown in Table 2, DeepLag achieves the best performance on Bounded Navier-Stokes, demonstrating its advanced ability to handle complex boundary conditions. In comparison to the previous best model, DeepLag achieves a significant 13.8% (Relative L2: 0.0618 v.s. 0.0544) and 2.7% (Relative L2: 0.1020 v.s. 0.0993) relative promotion on short and long rollout. The timewise error curves of all the models are also included in Figure 4. We can find that DeepLag presents slower error growth and excels in long-term forecasting. This result may stem from the Lagrangian-guided fluid prediction, which can accurately capture the dynamics information over time, further verifying the effectiveness of our design.

Table 2: Performance comparison on Bounded Navier-Stokes. Relative L2 of 10 frames and 30 frames prediction are recorded. Promotion represents the relative promotion of DeepLag w.r.t the second-best (underlined). "NaN" refers to the instability during rollout.

| Model | Relative L2 ($\downarrow$) | |
|---|---|---|
| | 10 Frames | 30 Frames |
| U-Net [32] | 0.0618 | 0.1038 |
| FNO [21] | 0.1041 | 0.1282 |
| Galerkin Transformer [3] | 0.1084 | 0.1369 |
| Vortex [7] | 0.1999 | NaN |
| GNOT [14] | 0.1388 | 0.1793 |
| LSM [52] | 0.0643 | 0.1020 |
| FactFormer [20] | 0.0733 | 0.1195 |
| DeepLag (Ours) | **0.0543** | **0.0993** |
| Promotion | 13.8% | 2.7% |

**Showcases**   To intuitively present the forecasting skills of different models, we also provide showcase comparisons in Figure 4 and particle movements predicted by DeepLag in Appendix K. We can find that DeepLag can precisely illustrate the vortex in the center of the figure and give a reasonable motion mode of the Kármán vortex phenomenon formed behind the upper left pillar. As for U-Net and LSM, although they successfully predicted the position of the center vortex, the error map shows that they failed to predict the density field as accurately as DeepLag. In addition, FactFormer deteriorates on this benchmark. This may be because it is based on spatial factorization, which is unsuitable for irregularly placed boundary conditions. These results further highlight the benefits of Eulerian-Lagrangian co-design, which can simultaneously help with dynamic and density prediction.

## 4.2 Ocean Current

**Setups**   Predicting large-scale ocean currents, especially in regions near tectonic plate boundaries prone to disasters such as tsunamis due to intense terrestrial activities, plays a crucial role in various domains. Hence, we also explore this challenging real-world scenario in our experiments. More details about the source and settings of this benchmark can be found in Appendix C.2.

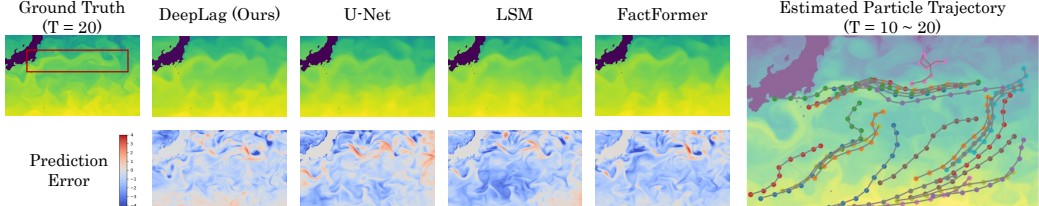

Figure 5: Showcase comparison and visualization of Lagrangian trajectories learned by DeepLag on Ocean Current. Notably, potential temperatures predicted by different models are plotted. Error maps of predictions are normalized to $(-4, 4)$ for a better view.

**Quantitive results** We report relative L2 for the Ocean Current dataset in Table 3, where DeepLag still achieves the best with 8.3% relative promotion w.r.t. the second-best model. Even in 30 days of forecasting, the number is 12.8%. These results show that DeepLag performs well in real-world, large-scale fluids, which usually involve more inherent stochasticity than simulated data. Moreover, we provide the ACC metric and timewise curve in Appendix F.

**Showcases** To provide an intuitive comparison, we plotted different model predictions in Figure 5. In comparison to other models, DeepLag exhibits the most minor prediction error. It accurately predicts the location of the high-temperature region to the south area and provides a clear depiction of the Kuroshio pattern [37] bounded by the red box.

**Learned trajectory visualization** To reveal the effect of learning Lagrangian trajectories, we visualize tracked particles in Figure 5. We observe that all the particles move from west to east, consistent with the Pacific circulation. Additionally, the tracked particles show distinct moving patterns, confirming their ability to represent complex dynamics. The move-

Table 3: Performance comparison on Ocean Current. We report the relative L2 of the short-term and long-term rollouts with their relative promotions.

| Model | Relative L2 (↓) | |
|---|---|---|
| | 10 Days | 30 Days |
| U-Net [32] | 0.0185 | 0.0297 |
| FNO [21] | 0.0246 | 0.0420 |
| Galerkin Transformer [3] | 0.0323 | 0.0515 |
| Vortex [7] | 0.9548 | NaN |
| GNOT [14] | 0.0206 | 0.0336 |
| LSM [52] | 0.0182 | 0.0290 |
| FactFormer [20] | 0.0183 | 0.0296 |
| DeepLag (Ours) | **0.0168** | **0.0257** |
| Promotion | 8.3% | 12.8% |

ment of upper particles matches the sinuous trajectory of the Kuroshio current, demonstrating the capability of DeepLag to provide interpretable evidence for prediction results. Visualizing the tracking points in Lagrangian space instills confidence in the reliability and interpretability of the predictions made by our model, which can provide valuable and intuitive insights for real-world fluid dynamics.

## 4.3 3D Smoke

**Setups** 3D fluid prediction has been a long-standing challenge due to the tanglesome dynamics involved. Therefore, we generated this benchmark to describe a scenario in which smoke flows under the influence of buoyancy in a three-dimensional bounding box. For more details, please refer to Appendix C.3.

**Quantitive results** Table 4 shows that DeepLag still performs best in 3D fluid. Note that in this benchmark, the canonical deep model U-Net degenerates seriously, indicating that a pure Eulerian multiscale framework is insufficient to model complex dynamics. We also noticed that the Transformer-based neural operators, such as

Table 4: Performance comparison on the 3D Smoke dataset. Relative L2 with relative promotion w.r.t. the second-best model is recorded.

| Model | Relative L2 (↓) |
|---|---|
| U-Net [32] | 0.0508 |
| FNO [21] | 0.0635 |
| Galerkin Transformer [3] | 0.1066 |
| GNOT [14] | 0.2100 |
| LSM [52] | 0.0527 |
| FactFormer [20] | 0.0793 |
| DeepLag (Ours) | **0.0378** |
| Promotion | 34.4% |

GNOT and Galerkin Transformer, failed in this task. This may be because both of them are based on the linear attention mechanism [19, 54], which may depreciate under massive tokens. These comparisons further highlight the capability of DeepLag to handle high-dimensional fluid.

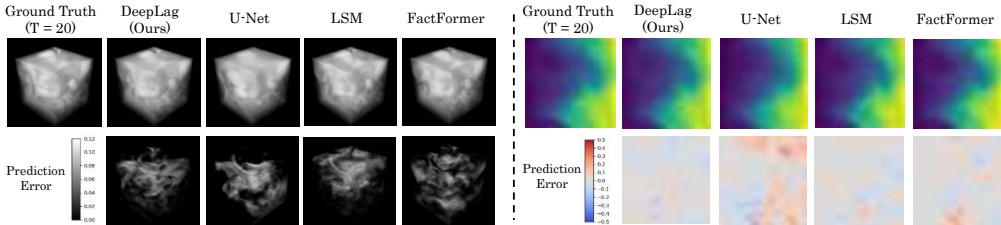

Figure 6: Showcases comparison of the whole space (left part) and a cross-section (right part) on the 3D Smoke dataset. For better visualization, we present the absolute value of the prediction error and normalize the whole space error into $(0, 0.12)$. As for the cross-section visualization, we choose the $xOy$ plane in the middle 3D fluid and normalize error maps to $(-0.5, 0.5)$.

**Showcases** In Figure 6, we compared the prediction results on the 3D Smoke dataset. DeepLag demonstrates superior performance in capturing both the convection and diffusion of smoke within the bounding box. In contrast, the predictions made by U-Net tend to average value across various surfaces, resulting in blurred details, which also indicates its deficiency in dynamics modeling. Similarly, LSM and FactFormer exhibit more pronounced errors, particularly around the smoke boundaries, where complex wave interactions often occur. By comparison, our model significantly reduces both the overall and cross-sectional error, excelling in the prediction of fine-grained, subtle flow patterns and maintaining high accuracy even in challenging regions with intricate dynamics.

### 4.4 Model Analysis

**Ablations** To verify the effectiveness of detailed designs in DeepLag, we conduct exhaustive ablations in Table 5. In our original design, we track 512 particles (around 3% of Eulerian grids) in total on 4 hierarchical spatial scales with a latent dimension 64 in the Lagrangian space.

The experiments indicate that further increasing the number of particles, scales, or latent dimensions yields marginal performance improvements. Therefore, we opt for these values to strike a balance between efficiency and performance. In addition, we can conclude that all components proposed in this paper are indispensable. Especially, the lack of interaction between the Eulerian and Lagrangian space will cause a severe drop in accuracy, highlighting the dual cross-attention that exploits Lagrangian dynamics has a positive influence on the evolution of Eulerian features. Besides, rather than uniformly sampling particles, sampling from a learnable probability matrix also provides an upgrade (refer to Appendix G for visual results). When swapping the

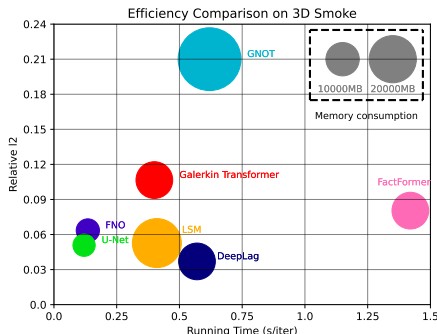

Figure 7: Efficiency comparison among all the models. Running time and Relative L2 are evaluated on the 3D Smoke benchmark.

Table 5: Ablations of **(a)** module removing and **(b)** hyperparameter sensitivity on Bounded Navier-Stokes, where (a) includes *w/o* LagToEu: removing Lagrangian-guided feature evolving, *w/o* EuToLag: removing Eulerian-conditioned particle tracking and *w/o Learnable Sampling*: removing learnable particle sampling strategy, and (b) includes *#Particle ($M_1$)*: adjusting the number of tracking particles of the first layer, *#Scale (L)*: adjusting the number of spatial scales and *#Latent ($C_1$)*: adjusting the number of latent dimension of the first layer. We use the control variable method, when one hyperparameter is changed, the values of the others are all original (*ori*). Ablations of **(c)** attention swapping on Bounded Navier-Stokes (*2D*) and 3D Smoke (*3D*), where the order of EuToLag and LagToEu blocks is swapped. *Original* and *Swapped* Relative L2 are reported.

| (a) Module Removing | | (b) Hyperparameter Sensitivity | | | | | | (c) Attention Swapping | | |
|---|---|---|---|---|---|---|---|---|---|---|

| Design | Relative L2 ($\downarrow$) | #Particle | Relative L2 ($\downarrow$) | #Scale | Relative L2 ($\downarrow$) | #Latent | Relative L2 ($\downarrow$) | Data | Original ($\downarrow$) | Swapped ($\downarrow$) |
|---|---|---|---|---|---|---|---|---|---|---|
| DeepLag | **0.0543** | 128 | 0.0559 | 1 | 0.0789 | 16 | 0.0656 | 2D | **0.0543** | 0.0545 |
| | | 256 | 0.0553 | 2 | 0.0658 | 32 | 0.0594 | 3D | 0.0378 | 0.0378 |
| w/o LagToEu($\cdot$) | 0.0556 | 512(ori) | **0.0543** | 4(ori) | **0.0543** | 64(ori) | **0.0543** | | | |
| w/o EuToLag($\cdot$) | 0.0547 | 768 | 0.0547 | 5 | 0.0554 | 128 | 0.0614 | | | |
| w/o Learnable Sampling | 0.0552 | | | | | | | | | |

Table 6: Experiments of **(a)** model efficiency alignment on 3D Smoke and **(b)** high-resolution data on Bounded Navier-Stokes. To compare with an aligned efficiency, we add more convolutional layers into the standard U-Net and increase the latent dimension. In the high-resolution simulation, we trained a new DeepLag model on a newly generated 256×256 Bounded Navier-Stokes dataset. Model parameters (*#Param*), GPU memory (*Mem*) and running time per epoch (*Time*) are reported.

| (a) Model Efficiency Alignment | | | | | (b) High-resolution Data | | | |
| --- | --- | --- | --- | --- | --- | --- | --- | --- |
| Model | #Param | Mem | Time | Relative L2 ($\downarrow$) | Resolution | Mem | Time | Relative L2 ($\downarrow$) |
| U-Net-scale | 336M | 13672MB | 601s/ep | NaN | $128 \times 128$ | 5420MB | 1150s/ep | 0.0543 |
| DeepLag | 19M | 12112MB | 845s/ep | **0.0378** | $256 \times 256$ | 13916MB | 1300s/ep | **0.0514** |

positions of the EuToLag and LagToEu blocks, the minimal performance change in both 2D and 3D benchmarks suggests the equivalence of the two perspectives, and the flow of information between them is bidirectional and insensitive to the order, underscoring the robustness of our approach. The above results provide solid support to our motivation in tracking essential particles and utilizing Eulerian-Lagrangian recurrence, further confirming the merits of our model.

**Efficiency analysis**  We also include the efficiency comparison in Figure 7. DeepLag hits a favorable balance between efficiency and performance by simultaneously considering performance, model parameters, and running time, demonstrating superior performance with significantly less memory than GNOT and LSM, thereby minimizing storage complexity. Standard U-Net has a large number of parameters, and Transformers have quadratic memory, so in large-scale or complex fluid prediction scenarios, using linear complexity attention mechanisms like ours is necessary. This explains why, although U-Net and LSM are good at Bounded Navier-Stokes, they will worsen when it comes to complex fluid dynamics, such as the Ocean Current and the 3D Smoke benchmark.

**Comparison under aligned efficiency**  As aforementioned, U-Net also presents competitive performance and efficiency in some cases. To carry out a fair comparison between baselines and DeepLag, we examine them on 3D Smoke, which scales up U-Net to have a running time similar to DeepLag as in Table 6(a). The results show that too many parameters overwhelm small baselines at the lower left corner in Figure 7 like U-Net, indicating that they have a shortcoming in scalability.

**Generalization analysis**  To verify the generalizing ability of DeepLag on larger and new domains, we ran tests on high-resolution (HR) data as in Table 6(b) and unseen boundary conditions (BC) as in Appendix J, respectively. The HR simulation on the $256 \times 256$ Bounded Navier-Stokes shows that the finer the data are, the more accurate our DeepLag is, which still has a 17% promotion w.r.t U-Net on relative L2 (DeepLag: 0.051, U-Net: 0.060). The comparison of GPU memory and running time per epoch also confirms that the increase of the space complexity is sublinear and the time complexity is minor, underscoring the scalability of DeepLag. For BCs, we ran a zero-shot test with the old model checkpoint on a newly generated Bounded N-S dataset, which has obstacles of different numbers, positions, and sizes. DeepLag still has a 7% promotion w.r.t the best baseline, U-Net, on relative L2 (DeepLag: 0.203, U-Net: 0.217). The visual comparison between the two models further shows that DeepLag adaptively generalizes well on unknown and more complex domains.

## 5   Conclusions and Limitations

To tackle intricate fluid dynamics, this paper presents DeepLag by introducing the Lagrangian dynamics into Eulerian fluid, which can provide clear and neat dynamics information for prediction. A EuLag Block is presented in the Eulerian-Lagrangian Recurrent Network to utilize the complementary advantages of Eulerian and Lagrangian perspectives, which brings better particle tracking and Eulerian fluid prediction. DeepLag excels in complex fluid prediction with an average improvement of nearly 10% across three carefully selected complex benchmarks, even in 3D fluid, and can also provide interpretable evidence by plotting learned Lagrangian trajectories. However, the number of tracking particles in DeepLag is appointed as a fixed hyperparameter that needs to be adjusted for specific scenarios, and missing Lagrangian supervision in the data leads to a lack of judgment on learned Lagrangian dynamics, which leave space for future exploration. That said, results show that the performance always tends to incline as we simply scale up. Therefore, the few hyperparameters for tuning depend more on the limitation of computing resources rather than blind searching.

## Acknowledgments and Disclosure of Funding

This work was supported by the National Natural Science Foundation of China (U2342217 and 62021002), the BNRist Project, and the National Engineering Research Center for Big Data Software.

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

# A  Implementation Details

This section provides the implementation details of DeepLag, including the configurations of model hyperparameters and the concrete design of modules.

## A.1  Hyperparameters

Detailed model configurations of DeepLag are listed in Table 7. Zero-padding is only used in the Ocean Current dataset to ensure the exact division in downsampling.

Table 7: Model configurations for DeepLag.

| Model Designs | Hyperparameters | Values |
|---|---|---|
| Eulerian-Lagrangian Recurrent Network | Number of observation steps $P$ | 10 |
| | Number of scales $L$ | 4 |
| | Sample Points at each scale $\{M_1, \cdots, M_L\}$ | $\{512, 128, 32, 8\}$ |
| | Downsample Ratio $r = \frac{\|\mathcal{D}_{l+1}\|}{\|\mathcal{D}_l\|}$ | 0.5 |
| | Channels of each scale $\{C_1, \cdots, C_L\}$ | $\{64, 128, 256, 256\}$ |
| | Paddings for Ocean Current dataset | $(12, 20)$ |
| EuLag Block | Heads in Cross-Attention | 8 |
| | Channels per head in Cross-Attention | 64 |

## A.2  Sampling and multiscale architecture in overall framework

### A.2.1  Sampling to initialize Lagrangian particles

At the first predicting step, the position of Lagrangian particles to track is initialized by the *dynamic sampling* module consisting of dynamics extracting and sampling.

**The** $\mathrm{ConvNet}(\cdot)$ **to extract dynamics**   Given the Eulerian input field $\{\mathbf{u}_t^l(\mathbf{x})\}_{\mathbf{x} \subset \mathcal{D}_l}$ at the $l$-th scale, the $\mathrm{ConvNet}(\cdot)$ operation in Eq. (4) is to extract the local dynamics around each Eulerian observation point with $\mathrm{Conv}()$, $\mathrm{BatchNorm}()$ and $\mathrm{ReLU}()$ layers, which can be formalized as follows:

$$\mathrm{ConvNet}(\mathbf{u}_t) = \mathrm{Conv}\left( \mathrm{ReLU}\left( \mathrm{BatchNorm}\left( \mathrm{Conv}(\{\mathbf{u}_t^l(\mathbf{x})\}_{\mathbf{x} \subset \mathcal{D}_l}) \right) \right) \right), \ l \text{ from 1 to } L. \quad (12)$$

Here, the output channel of the outermost $\mathrm{Conv}$ is 1.

**The** $\mathrm{Sample}(\cdot)$ **to select key particles**   Given the probability distribution matrix $\mathbf{S}_t^l(\mathbf{x})_{\mathbf{x} \subset \mathcal{D}_l}$ and the number of particles to sample $M_l$ at the $l$-th scale, we choose the particles for further tracking by $\mathrm{Multinomial}(\cdot)$ without replacement:

$$\mathrm{Sample}(\mathbf{S}_t^l) = \mathrm{Multinomial}(\mathbf{S}_t^l, M_l), \ l \text{ from 1 to } L. \quad (13)$$

### A.2.2  Multiscale architecture

Multiscale modeling is utilized in DeepLag as represented in Figure 2(d), where we need to maintain multiscale deep Eulerian features as follows:

**Encoder**   Given Eulerian fluid observation $\{\mathbf{u}_t(\mathbf{x})\}_{\mathbf{x} \subset \mathcal{D}}$ at the $t$-th time step, $\{\mathbf{u}_{(t-P+1):(t-1)}(\mathbf{x})\}_{\mathbf{x} \subset \mathcal{D}}$ at previous time steps and the 0-1 boundary-geometry mask $\{\boldsymbol{m}(\mathbf{x})\}_{\mathbf{x} \subset \mathcal{D}}$ where 1 indicates the border and the unreachable area (like pillars in Bounded Navier-Stokes), the $\mathrm{Encode}()$ operation is to project original fluid properties in physical domain to deep representations with linear layer and position embedding, which can be formalized as follows:

$$\mathbf{u}_t^1 = \mathrm{Linear}\left( \mathrm{Concat}\left( \{\mathbf{u}_{(t-P+1):t}(\mathbf{x}), \boldsymbol{m}(\mathbf{x})\}_{\mathbf{x} \subset \mathcal{D}} \right) \right) + \mathrm{PosEmbedding}. \quad (14)$$

For position embedding, we concatenate two additional channels to the input (three for 3D Smoke), representing normalized $(x, y)$ coordinates (or $(x, y, z)$ for 3D Smoke).

**Decoder**   Given evolved Eulerian deep representations in the finest scale $\{\mathbf{u}^1_{t+1}(\mathbf{x})\}_{\mathbf{x} \subset \mathcal{D}}$ at the $(t+1)$-th time step, the Decode() operation is to project deep representations back to predicted fluid properties with two linear layers and a GeLU activation [16], which can be formalized as follows:

$$\mathbf{u}_{t+1} = \text{Linear}\Big(\text{GeLU}\big(\text{Linear}(\{\mathbf{u}^1_{t+1}(\mathbf{x})\}_{\mathbf{x} \subset \mathcal{D}})\big)\Big). \tag{15}$$

**Downsample**   Given Eulerian deep representations $\{\mathbf{u}^l_t(\mathbf{x})\}_{\mathbf{x} \subset \mathcal{D}_l}$ at the $l$-th scale, the Down() operation is to concentrate local information of deep representations into a smaller feature map at the $(l+1)$-th scale with MaxPooling() and Conv() layers, which can be formalized as follows:

$$\mathbf{u}^{l+1}_t = \text{Conv}\Big(\text{MaxPooling}\big(\{\mathbf{u}^l_t(\mathbf{x})\}_{\mathbf{x} \subset \mathcal{D}_l}\big)\Big), \ l \text{ from } 1 \text{ to } (L-1). \tag{16}$$

**Upsample**   Given the evolved Eulerian deep representations $\{\mathbf{u}^l_{t+1}(\mathbf{x})\}_{\mathbf{x} \subset \mathcal{D}_l}$ at the $l$-th scale and $\{\mathbf{u}^{l+1}_{t+1}(\mathbf{x})\}_{\mathbf{x} \subset \mathcal{D}_{l+1}}$ at the $(l+1)$-th scale, respectively, the Up() operation is to fuse information on corresponding position between two adjacent scales of deep representations into a feature map at the $l$-th scale with Interpolate() operation and Conv() layers, which can be formalized as follows:

$$\mathbf{u}^l_{t+1} = \text{Conv}\left(\text{Concat}\left(\Big[\text{Interpolate}\big(\{\mathbf{u}^{l+1}_{t+1}(\mathbf{x})\}_{\mathbf{x} \subset \mathcal{D}_{l+1}}\big), \{\mathbf{u}^l_{t+1}(\mathbf{x})\}_{\mathbf{x} \subset \mathcal{D}_l}\Big]\right)\right), \ l \text{ from } (L-1) \text{ to } 1 \tag{17}$$

## A.3   EuLag Block

**The** LagToEu($\cdot$) **process**   Hereafter we denote by $\mathbf{h}_t || \mathbf{p}_t$ for the concatenated representations. As we stated in subsection 3.2, the Lagrangian-guided Eulerian feature evolving process, short as LagToEu($\cdot$), aggregates information from Lagrangian description to guide the update of Eulerian field with a single Transformer layer at each scale:

$$\begin{aligned}
\mathbf{u}^l_{t+1} &= \mathbf{u}^l_t + \text{LagToEu}(\mathbf{u}^l_t, \mathbf{h}^l_t || \mathbf{p}^l_t, \mathbf{h}^l_t || \mathbf{p}^l_t) \\
&= \mathbf{u}^l_t + \text{FFN}(\text{LagToEu-Attn}(\mathbf{u}^l_t, \mathbf{h}^l_t || \mathbf{p}^l_t, \mathbf{h}^l_t || \mathbf{p}^l_t))) + \text{LagToEu-Attn}(\mathbf{u}^l_t, \mathbf{h}^l_t || \mathbf{p}^l_t, \mathbf{h}^l_t || \mathbf{p}^l_t),
\end{aligned} \tag{18}$$

where LagToEu-Attn($\mathbf{u}^l_t, \mathbf{h}^l_t || \mathbf{p}^l_t, \mathbf{h}^l_t || \mathbf{p}^l_t$) is described as Eq. (7), $l \in \{1, 2, \ldots, L\}$. Moreover, we use the pre-normalization [53] technique for numerical stability.

**The** EuToLag($\cdot$) **process**   Similar to above, we acquire the new particle position $\mathbf{p}^l_{t+1}$ and the global dynamics $\mathbf{h}^l_{\text{global},t+1}$ in the Eulerian-conditioned particle tracking by a Eulerian-Lagrangian cross-attention:

$$\begin{aligned}
\mathbf{h}^l_{\text{global},t+1} || \mathbf{p}^l_{t+1} &= \mathbf{h}^l_t || \mathbf{p}^l_t + \text{EuToLag}(\mathbf{h}^l_t || \mathbf{p}^l_t, \mathbf{u}^l_{t+1}, \mathbf{u}^l_{t+1}) \\
&= \mathbf{h}^l_t || \mathbf{p}^l_t + \text{FFN}(\text{EuToLag-Attn}(\mathbf{h}^l_t || \mathbf{p}^l_t, \mathbf{u}^l_{t+1}, \mathbf{u}^l_{t+1})) \\
&\quad + \text{EuToLag-Attn}(\mathbf{h}^l_t || \mathbf{p}^l_t, \mathbf{u}^l_{t+1}, \mathbf{u}^l_{t+1}),
\end{aligned} \tag{19}$$

where $l \in \{1, 2, \cdots L\}$. Then, we interpolate from $\mathbf{u}^l_{t+1}$ by $\mathbf{p}^l_{t+1}$ to be the local dynamics $\mathbf{h}^l_{\text{local},t+1}$ and Aggregate it with the global dynamics $\mathbf{h}^l_{\text{global},t+1}$ using a Linear layer:

$$\begin{aligned}
\mathbf{h}_{t+1} &= \text{Aggregate}\big(\text{Interpolate}(\mathbf{u}_{t+1}, \mathbf{p}_{t+1}), \mathbf{h}_{\text{global},t+1}\big) \\
&= \text{Linear}\Big(\text{Concat}\big(\mathbf{h}^l_{\text{global},t+1}, \text{Interpolate}(\mathbf{u}^l_{t+1}, \mathbf{p}^l_{t+1})\big)\Big).
\end{aligned} \tag{20}$$

## A.4   Metrics

**Relative L2**   We use the relative L2 as the primary metric for all three tasks. Compared to MSE, Relative L2 is less influenced by outliers and is more robust. For given $n$ steps 2D predictions $\hat{\mathbf{x}} \in \mathbb{R}^{H \times W \times n}$ or 3D predictions $\hat{\mathbf{x}} \in \mathbb{R}^{H \times W \times C \times n}$ and their corresponding ground truth $\mathbf{x}$ of the same size, the relative L2 can be expressed as:

$$\text{Relative L2} = \frac{\|\mathbf{x} - \hat{\mathbf{x}}\|_2^2}{\|\mathbf{x}\|_2^2}, \tag{21}$$

where $\| \cdot \|_2$ represents the L2 norm.

# B Comparison Between DeepLag and Classical ML Methods

DeepLag differs significantly from classical ML approaches like Neural ODE [4], as highlighted in the comparison Table 8:

Table 8: Comparison between DeepLag and classical ML model

| Feature | DeepLag | Neural ODE |
|---|---|---|
| ODE Specification | Not required | Explicitly specified |
| Computational Paradigm | Data-driven | Model-driven |
| Attention Mechanism | Utilizes attention | Typically not utilized |
| Integration of Lagrangian Dynamics | Integrated | Operates solely in Euler space |
| Input-Output Mapping | Complex, high-dimensional PDE | Simple, single-variable ODE |

While Neural ODE methods explicitly specify ODEs, fluid dynamics is governed by multi-variable PDEs, rendering the ODE formulation inadequate. Moreover, DeepLag is data-driven and does not require explicit ODE specification. Notably, we are the first to employ attention mechanisms for computing Lagrangian dynamics, which closely resemble operators in both deep learning and numerical Lagrangian methods. Additionally, DeepLag integrates both Eulerian and Lagrangian frameworks, whereas ODE-based methods operate solely in the Eulerian space. Finally, DeepLag models the complex mapping of high-dimensional spatiotemporal PDEs, which is significantly more intricate than the simpler processes modeled by single-variable (either temporal or spatial) ODEs.

# C More Details about the Benchmarks

## C.1 Bounded Navier-Stokes

Here we provide some details about the benchmark Bounded Navier-Stokes. Just as its name implies, the motion of dye is simulated from the Navier-Stokes equation of incompressible fluid. 2000 sequences with spatial resolution of $128 \times 128$ are generated for training and 200 new sequences are used for the test. We supplement important details indicating the difficulty of the Bounded Navier-Stokes dataset as follows:

**About the Reynolds number** The Reynolds number of the dataset is 256. At this Reynolds number, attached vortices dissipate and form a boundary layer separation. The downstream flow field behind the cylinder becomes unsteady, with vortices shedding periodically on both sides of the cylinder's rear edge, resulting in the well-known phenomenon of *Kármán vortex street* [50]. Additionally, due to the presence of multiple cylinders within the flow field and obstruction from downstream cylinders, more complex flow phenomena than flow around a cylinder occur, challenging the model's capacity.

**Differences of data sequences** Our data generation method involves running simulations for over $10^5$ steps after setting the initial conditions of the flow field. We then randomly select a starting time step and extract several frames as an example, which are further randomly divided into training and testing sets. The positions of the cylinders are fixed, but the initial condition varies in different samples, which can simulate a scenario like bridge pillars in a torrential river. Due to the highly unsteady nature of the flow field, the flow patterns observed by the model appear significantly different.

**Numerical method used in data generation** We utilized the Finite Difference Method (MAC Method) with CIP (Constrained Interpolation Profile) as the Advection Scheme for numerical simulations, implemented by [36]. This high-order interpolation-constrained format effectively reduces numerical dissipation, enhancing the accuracy and reliability of numerical simulations.

## C.2 Ocean Current

Some important details related to the Ocean Current benchmark are attached here. Learning ocean current patterns from data and providing long-term forecasts are of great significance for disaster prevention and mitigation, which motivates us to focus on this problem.

The procedures to make this dataset are as follows. First, we downloaded daily sea reanalysis data [5] from 2011 to 2020 provided by the ECMWF and selected five basic variables on the sea surface to construct the dataset, including velocity, salinity, potential temperature, and height above the geoid, which are necessary to identify the ocean state. Then, we crop a $180 \times 300$ sub-area on the North Pacific from the global record, corresponding to a 375km×625km region. In total, this dataset consists of 3,653 frames, where the first 3000 frames are used for training, and the last 600 frames are used for testing. The training task is to predict the future current of 10 days based on the past 10 days' observation, after which we performed 30 days of inference with the trained model to examine the long-term stability of DeepLag.

### C.3 3D Smoke

To verify our model effectiveness in this complex setting of the high-dimensional tanglesome molecular interaction, we also generate a 3D fluid dataset for the experiment. This benchmark consists of a scenario where smoke flows under the influence of buoyancy in a three-dimensional bounding box. This process is governed by the incompressible Navier-Stokes equation and the advection equation of fluid. 1000 sequences are generated for training, and 200 new samples are used for testing. Each case is in the resolution of $32^3$.

## D Training Curves

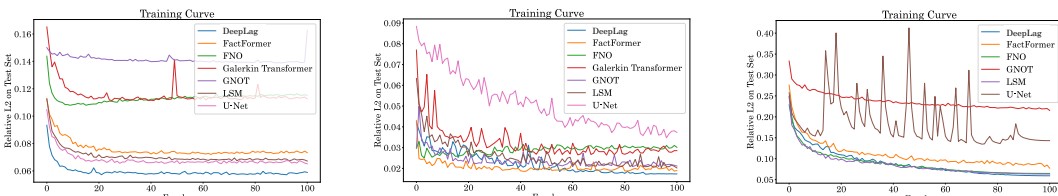

Figure 8: Training curve comparison among all the models on Bounded Navier-Stokes dataset, Ocean Current, and 3D Smoke dataset.

We provide training curves on Bounded Navier-Stokes, Ocean Current, and 3D Smoke datasets in Figure 8. We can observe that DeepLag presents favorable training robustness and converges the fastest on the Bounded Navier-Stokes dataset.

## E Analysis on the Parameter Count and Performance Difference

In Table 1 in our paper, we present three different datasets with variations in data type, number of variables, number of dimensions, and spatial resolution. Specifically, the 2D fluid and 3D fluid represent *entirely different dynamic systems*, so it is normal for different baselines to perform inconsistently across different benchmarks. In the following results, `patch_size` of all models are set to 1 for fair comparison. For instance, U-Net performs well on datasets with *distinct multiscale attributes* (such as the Bounded Navier-Stokes dataset), while transformer-based methods excel on datasets with *broad practical ranges* (like the Ocean Current dataset) *or high dimensions* (such as the 3D Smoke dataset), where attention mechanisms can effectively model global features. Hence, our model's ability to handle multiscale and global modeling simultaneously *highlights the challenge of achieving consistent state-of-the-art performance*. Below are the parameter statistics for DeepLag and all baseline models across each benchmark, along with some experiments we conducted to ensure parity in the parameter count for each model.

### E.1 Analysis on Bounded Navier-Stokes benchmark

The parameter quantities for each model on the Bounded Navier-Stokes benchmark are shown in Table 9. When attempting to further increase the parameter count of smaller models for fair comparison, we encountered CUDA Out-of-memory errors. Specifically, these errors occurred when we increased the `Dynamic_net_layer` of Vortex from 4 to 50, resulting in a parameter count of 171,761, but encountered CUDA Out of Memory issues.

Table 9: Model parameter summary for Bounded Navier-Stokes dataset.

| Model | #Parameter |
| --- | --- |
| UNet [32] | 17,311,489 |
| FNO [21] | 4,744,513 |
| Galerkin-Transformer [3] | 3,917,600 |
| Vortex [7] | 96,321 |
| GNOT [14] | 3,736,713 |
| LSM [52] | 19,188,033 |
| FactFormer [20] | 5,477,185 |
| DeepLag (Ours) | 10,895,771 |

## E.2 Analysis on Ocean Current benchmark

Table 10: Model parameter summary for Ocean Current dataset.

| Model | #Parameter |
| --- | --- |
| UNet [32] | 17,314,565 |
| FNO [21] | 4,747,589 |
| Galerkin-Transformer [3] | 3,941,156 |
| GNOT [14] | 993,101 |
| Vortex [7] | 96,321 |
| LSM [52] | 19,191,109 |
| FactFormer [20] | 5,482,821 |
| DeepLag (Ours) | 13,782,431 |

The parameter quantities for each model on the Ocean Current dataset are summarized in Table 10. When attempting to further increase the parameter count of smaller models for fair comparison, we encountered CUDA Out-of-memory errors. Specifically, these errors occurred when we increased the `Dynamic_net_layer` of Vortex from 4 to 50, resulting in a parameter count of 171,761, but encountered CUDA Out of Memory issues. Similarly, for GNOT, increasing the `latent_dim` from 64 to 96 resulted in a parameter count of 1,659,179, yet this configuration encountered CUDA Out of Memory errors.

## E.3 Analysis on 3D Smoke benchmark

Table 11: Model parameter summary for 3D Smoke dataset.

| Model | #Parameter |
| --- | --- |
| UNet [32] | 51,892,292 |
| FNO [21] | 56,651,908 |
| Galerkin-Transformer [3] | 28,867,841 |
| GNOT [14] | 1,450,768 |
| LSM [52] | 25,937,732 |
| FactFormer [20] | 1,840,004 |
| DeepLag (Ours) | 19,526,827 |

The parameter quantities for each model on the 3D Smoke dataset are shown in Table 11. When experimenting with increasing the parameter count of certain models for fair comparison, we encountered CUDA Out-of-memory errors. Specifically, these errors occurred when we increased the `latent_dim` of GNOT from 64 to 96, resulting in a parameter count of 1,450,768 for GNOT, which encountered CUDA Out of Memory issues. Similarly, when adjusting the `encoder_transformer_layer` of FactFormer from 3 to 13, the parameter count reached 7,382,084, yet this configuration encountered CUDA Out of Memory errors.

Figure 7 in our paper, along with the provided tables above, illustrate our rigorous comparison and efforts to standardize model parameters for a fair comparison. However, due to excessive GPU

Table 12: ACC results on the Ocean Current dataset and the curve of timewise ACC. Both ACC averaged from 10 prediction steps and ACC of the last prediction frame are recorded. A higher ACC value indicates better performance. Relative promotion is also calculated.

| MODEL | AVG. ACC (↑) | LAST. ACC (↑) |
|---|---|---|
| U-NET [32] | 0.0916 | 0.0888 |
| FNO [21] | 0.0883 | 0.0853 |
| GALERKIN TRANSFORMER [3] | 0.0896 | 0.0848 |
| GNOT [14] | 0.4725 | 0.4557 |
| LSM [52] | 0.2378 | 0.2305 |
| FACTFORMER [20] | 0.4789 | 0.4633 |
| DEEPLAG (OURS) | **0.4820** | **0.4694** |
| PROMOTION | 0.6% | 1.3% |

Timewise ACC (top 3)

memory consumption of intermediate results in some baseline models, we could not conduct direct performance comparisons under matched parameter conditions. The ability of our model to efficiently utilize GPU memory resources is a valuable aspect of practical applications.

## F  ACC Metric on Ocean Current

**Latitude-weighted Anomaly Correlation Coefficient**  In meteorology, directly calculating the correlation between predictions and ground truth may obtain misleadingly high values because of the seasonal variations. To subtract the climate average from both the forecast and the ground truth, we utilize the Anomaly Correlation Coefficient to verify the forecast and observations. Moreover, since the observation grids are equally spaced in longitude, and the size of the different grids is related to the latitude, we calculate the latitude-weighted Anomaly Correlation Coefficient, which can be formalized as:

$$\text{ACC}(v, t) = \frac{\sum_{i,j} \text{Lat}(\phi_i) \hat{\mathbf{x}}'^v_{i,j,t} \mathbf{x}'^v_{i,j,t}}{\sqrt{\sum_{i,j} \text{Lat}(\phi_i) \left(\hat{\mathbf{x}}'^v_{i,j,t}\right)^2 \times \sum_{i,j} \text{Lat}(\phi_i) \left(\mathbf{x}'^v_{i,j,t}\right)^2}}, \tag{22}$$

where $v$ represents a certain observed variable, $\hat{\mathbf{x}}_{i,j,t}$ is the prediction of ground truth $\mathbf{x}$ at position $i, j$ and forecast time $t$. $\mathbf{x}' = \mathbf{x} - \bar{\mathbf{x}}$ represents the difference between $\mathbf{x}$ and the climatology $\bar{\mathbf{x}}$, that is, the long-term mean of observations in the dataset. $\text{Lat}(\phi_i) = N_{\text{Lat}} \times \frac{\cos \phi_i}{\sum_{i'=1}^{N_{\text{Lat}}} \cos \phi_{i'}}$, where $N_{\text{Lat}} = 180$ and $\phi_i$ is the latitude of the $i$-th row of output.

**ACC result on the Ocean Current benchmark**  Notably, DeepLag also excels in the ACC metric, as shown in Table 12, which can better quantify the model prediction skill. As shown in the timewise ACC curve, DeepLag consistently achieves the highest ACC and holds more significant advantages in long-term prediction. Since ACC is calculated with long-term climate statistics, further improvements become increasingly challenging as it increases, which highlights the value of DeepLag.

## G  Visual Results for Learnable Sampling

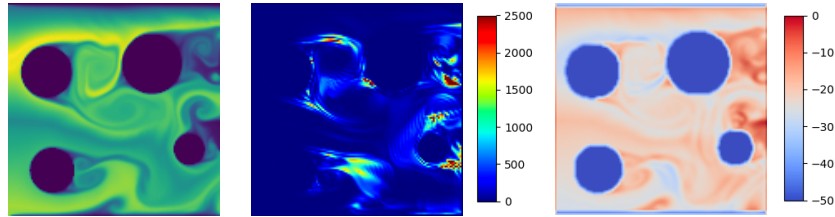

Figure 9: Visualization of the pointwise variance of vorticity (middle) and the sampling probability distribution (right) learned by DeepLag. We plot the $\log(\mathbf{S})$ here for a better view.

To demonstrate the effectiveness of our learnable probability, we visualize its distribution $\mathbf{S}$ with respect to the pointwise variance of vorticity, which is directly proportional to the local complexity of

fluid dynamics, in Figure 9. It is evident that our sampling module tends to prioritize focusing on and sampling particles within regions having higher dynamical complexity, such as the *wake flow* and *Karman vortex* [50] near the domain borders and behind the pillar. The showcase in Figure 1 also demonstrates that the tracked particle can well present the dynamics of a certain area. This observation underscores that our design is very flexible at adapting to various complex boundary conditions and can effectively guide the model to track the most crucial particles, enhancing its performance in capturing the fine details in the wake zone where turbulence and vortex form.

## H   More Showcases

As a supplement to the main text, we provide more showcases here for comparison (Figure 10-15).

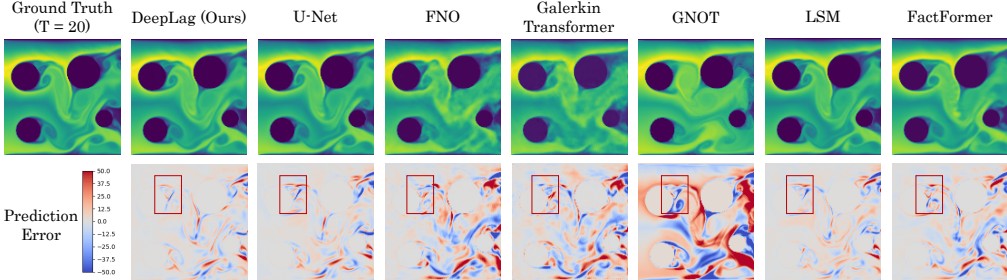

Figure 10: Showcases of the Bounded Navier-Stokes dataset.

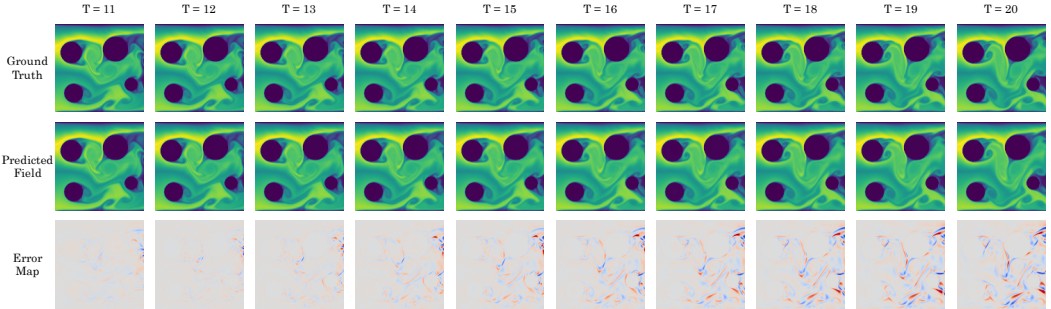

Figure 11: Showcases of DeepLag on the Bounded Navier-Stokes dataset.

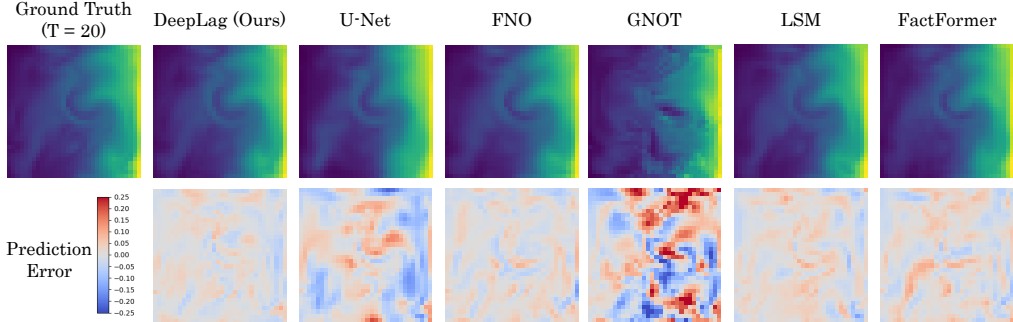

Figure 12: Showcases of the 3D Smoke dataset.

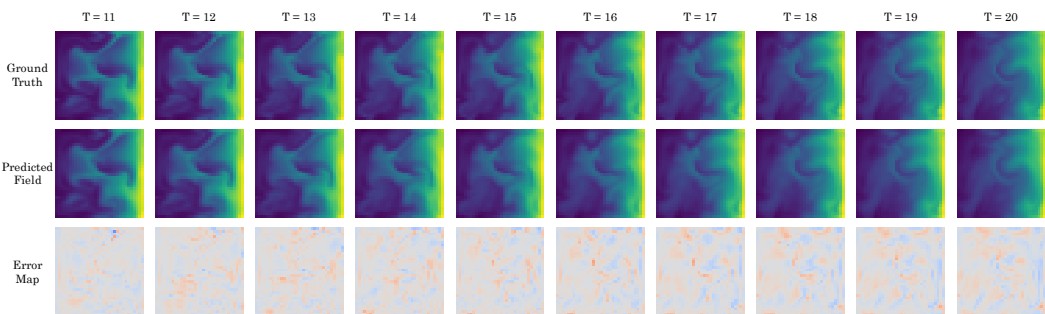

Figure 13: Showcases of DeepLag on the 3D Smoke dataset.

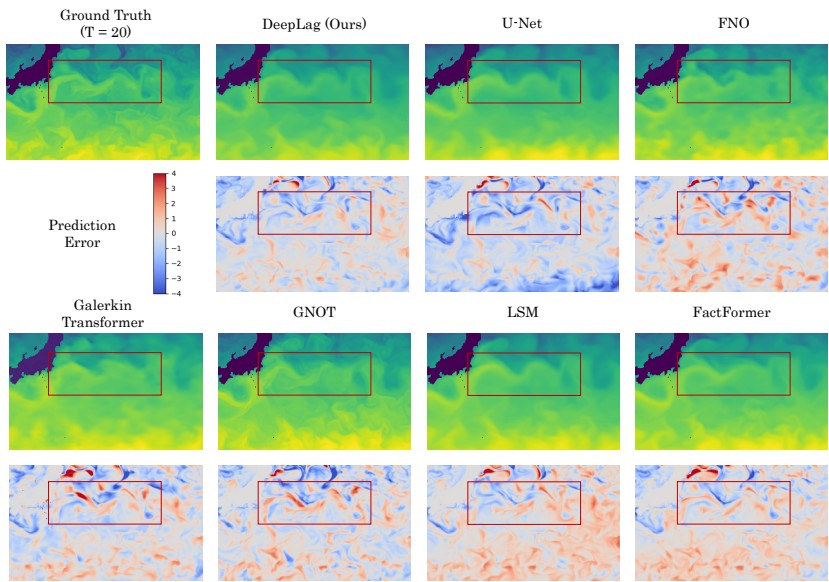

Figure 14: Showcases of the Ocean Current dataset.

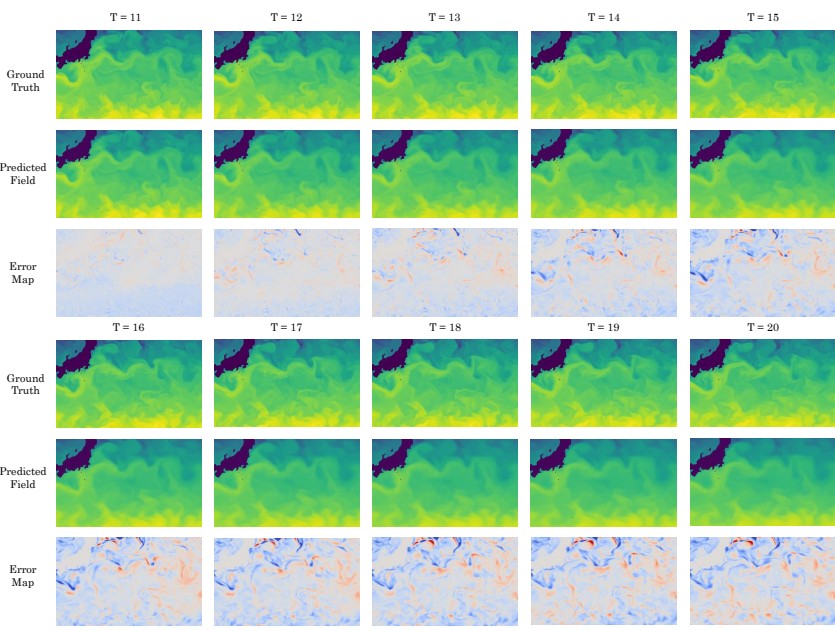

Figure 15: Showcases of DeepLag on the Ocean Current dataset.

# I Result of Long-term Prediction

## I.1 Reason for predicting 10 steps

Predicting the last 10 steps from the first 10 steps input is a convention (refer to FNO where $\nu = 1e - 5, T = 20$ is the setting, and later several baselines using the NS dataset also followed this setting). We did this to follow the convention and for ease of comparison. Additionally, the Ocean Current dataset has a one-day interval between every two frames, and predicting 10 days ahead is already a long horizon.

## I.2 Results of long-term rollout

We conducted experiments for long-term (extrapolation beyond hundreds of frames) predictions. Accurately, we utilize trained 10-frame prediction models to perform 100-frame prediction. Our reason for not directly training a model in an autoregressive paradigm to predict the next 100 frames is due to insufficient memory capacity and the issue of gradient explosion or vanishing. The results are as follows. The best result is bold and the second is underlined. The reason we did not run the 3D Smoke dataset is that it is very challenging to load 110 frames of large 3D data at once, which overwhelms our machine. Extra video results are in supplementary materials, which effectively illustrate the performance and consistency of our model's predictions.

### I.2.1 Bounded Navier-Stokes

**Quantitive results** As depicted in Table13, the DeepLag model still outperforms the strong baselines on the Bounded Navier-Stokes dataset in predicting 100 frames into the future. With a Relative L2 of 0.1493, DeepLag achieves superior performance compared to all baselines. Additionally, the performance trends of all models are visually illustrated in Figure 16-20, revealing DeepLag's ability to maintain lower error growth rates over time, particularly in long-term predictions. This outcome suggests that the Lagrangian particle-based approach adopted by DeepLag effectively captures dynamic information, contributing to its robust forecasting capability in fluid dynamics modeling.

Table 13: Performance comparison for predicting 100 frames on the Bounded Navier-Stokes dataset. Relative L2 is recorded. For clarity, the best result is in bold and the second best is underlined. Promotion represents the relative promotion of our model w.r.t the second best model.

| Model | Relative L2 ($\downarrow$) |
|---|---|
| U-Net [32] | 0.1529 |
| FNO [21] | 0.4244 |
| Galerkin-Transformer [3] | 0.3010 |
| GNOT [14] | 0.2489 |
| LSM [52] | 0.1511 |
| FactFormer [20] | 0.1691 |
| DeepLag (Ours) | **0.1493** |
| promotion | 1.2% |

**Showcases** To visually evaluate the predictive capabilities of our models on long-term, we present a showcase of last frame comparisons and time-wise prediction with error map in Figure 16-20, illustrating the long-term rollout performance on the Bounded Navier-Stokes dataset. Notably, DeepLag demonstrates remarkable accuracy in capturing complex flow phenomena, accurately predicting the formation and evolution of vortices, particularly the Kármán vortex street behind the upper left pillar. In contrast, U-Net and LSM exhibit moderate success in predicting the central vortex but struggle with accurately reproducing the density distribution of the flow field, as indicated by the error maps. FactFormer, however, shows subpar performance on this benchmark, likely due to its reliance on spatial factorization, which may not effectively handle irregular boundary conditions. These findings underscore the advantages of our Eulerian-Lagrangian co-design approach, which enables simultaneous prediction of dynamics and density, contributing to more accurate and comprehensive fluid modeling and forecasting capabilities.

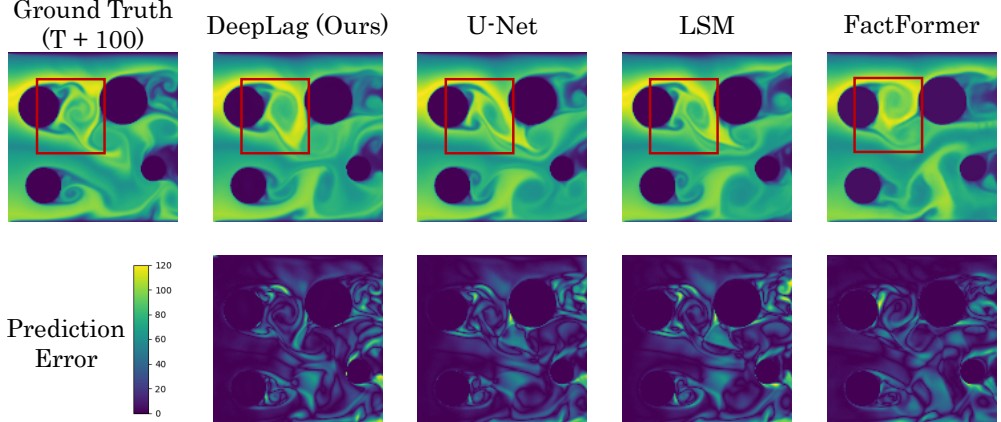

Figure 16: Showcases comparison between the most competitive models of long-term rollout on the Bounded Navier-Stokes benchmark.

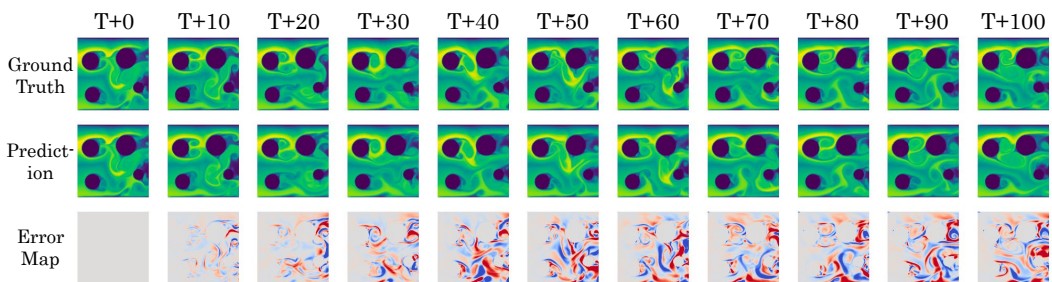

Figure 17: Timewise showcases of DeepLag of long-term rollout on the Bounded Navier-Stokes benchmark.

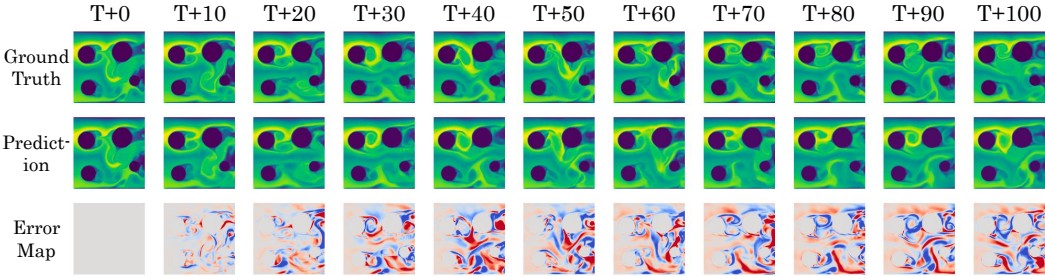

Figure 18: Timewise showcases of FactFormer of the long-term rollout on the Bounded Navier-Stokes benchmark.

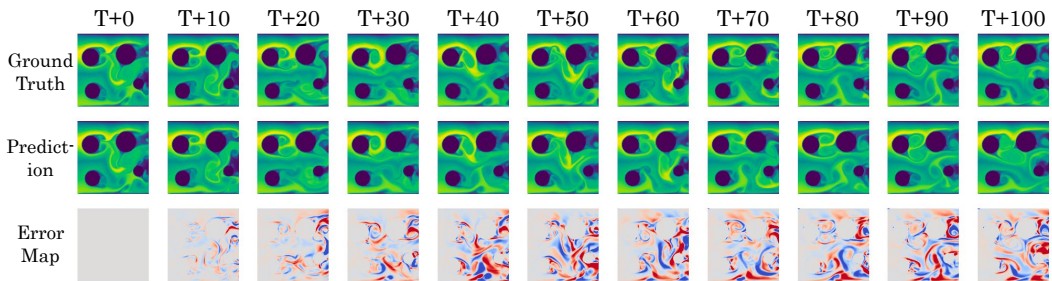

Figure 19: Timewise showcases of LSM of the long-term rollout on the Bounded Navier-Stokes benchmark.

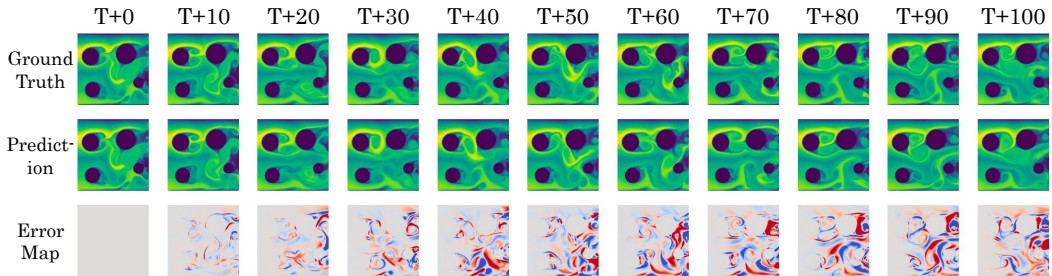

Figure 20: Timewise showcases of U-Net of long-term rollout on the Bounded Navier-Stokes benchmark.

### I.2.2 Ocean Current

**Quantitive results** We present a comparison of results for the Ocean Current dataset in Table 14, which includes relative L2, Last frame ACC, and Average ACC metrics. DeepLag maintains its superiority, achieving the lowest relative L2 among all models, with an 8.7% promotion compared to the second-best model. These findings highlight DeepLag's robust performance in predicting real-world large-scale fluid dynamics, which often exhibit inherent stochasticity. Furthermore, DeepLag outperforms other models in ACC metrics, indicating its superior predictive capability. This is further corroborated by the timewise ACC curve, where DeepLag consistently demonstrates the highest ACC values, particularly in long-term predictions.

**Showcases** To visually assess the long-term forecasting performance of each model, we showcase the last frame predictions along with their errors, and the time-wise prediction errors for each model in Figure 21-25. Visually, our DeepLag predictions closely resemble the Ground Truth compared to other models, demonstrating robust long-term extrapolation capabilities and accurate capture of the *Kuroshio pattern* [37]. It can be observed that FactFormer and LSM exhibit relatively large errors, while GNOT tends to average and loses fine texture details. However, DeepLag does not suffer from these issues.

### I.3 Examination on the turbulent kinetic energy spectrum

In the field of fluid mechanics, simulation results that better adhere to intrinsic physical laws are sometimes more valuable than those with smaller pointwise errors, often reflected in frequency domain analysis. To validate this point and to measure long-term forecasting ability, we introduced a metric on time-averaged turbulent statistics, namely the turbulent kinetic energy spectrum (TKES). Specifically, we computed the MAE and RMSE of TKES for the Bounded Navier-Stokes dataset, which exhibits the most prominent turbulent characteristics, and plotted the line graphs of the error of

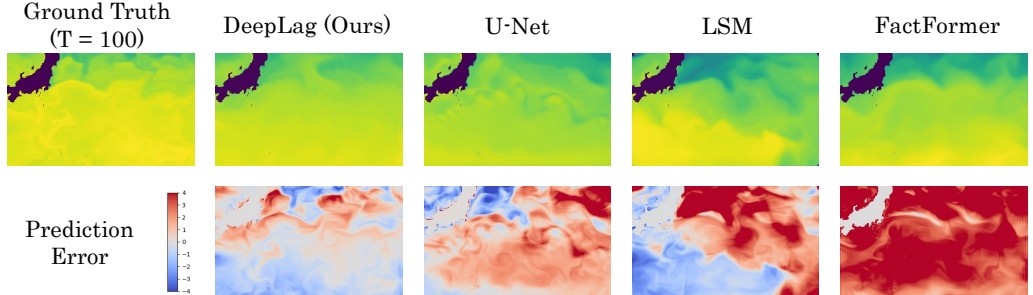

Figure 21: Showcases comparison between the most competitive models of the long-term rollout on the Ocean Current benchmark.

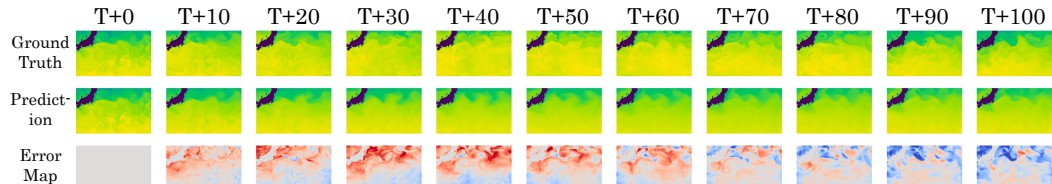

Figure 22: Timewise showcases of DeepLag of the long-term rollout on the Ocean Current benchmark.

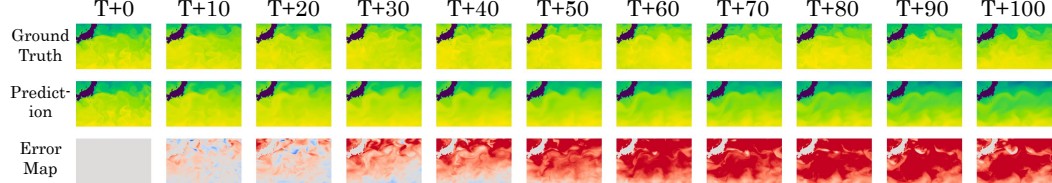

Figure 23: Timewise showcases of FactFormer of the long-term rollout on the Ocean Current benchmark.

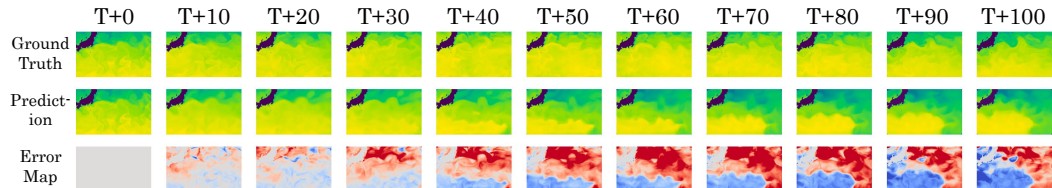

Figure 24: Timewise showcases of LSM of the long-term rollout on the Ocean Current benchmark.

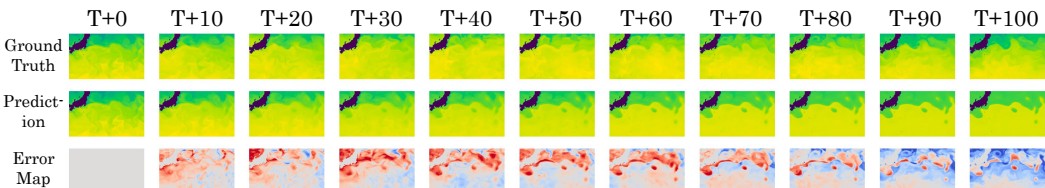

Figure 25: Timewise showcases of GNOT of the long-term rollout on the Bounded Navier-Stokes benchmark.

Table 14: Relative L2 for predicting 100 frames, Last frame ACC, and Average ACC

| Model | Relative L2 ($\downarrow$) | Last frame ACC ($\uparrow$) | Average ACC ($\uparrow$) |
|---|---|---|---|
| U-Net [32] | 0.0764 | 0.0197 | 0.0503 |
| FNO [21] | 0.1675 | 0.0041 | 0.0448 |
| Galerkin-Transformer [3] | NaN | NaN | NaN |
| GNOT [14] | 0.0513 | 0.173 | 0.3586 |
| LSM [52] | 0.0616 | 0.0793 | 0.1650 |
| FactFormer [20] | 0.0460 | 0.2647 | 0.3864 |
| DeepLag (Ours) | **0.0423** | **0.2890** | **0.4041** |
| promotion | 8.7% | 9.2% | 4.6% |

wave number and TKE, as shown in Table 15. It can be observed both numerically and visually that DeepLag consistently outperforms various baselines to different extents.

Table 15: Turbulent kinetic energy spectrum (TKES) results on the Bounded Navier-Stokes. Both the plot of error of TKES (left) and MAE and RMSE of TKES (right) are presented. A lower TKES error value indicates better performance. Relative promotion is also calculated.

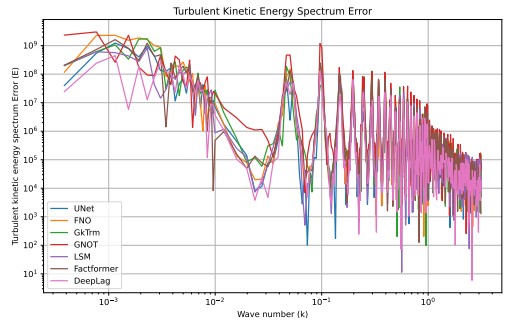

| MODEL | MAE ($\downarrow$) | RMSE ($\downarrow$) |
|---|---|---|
| U-NET | 3.888E+06 | 3.349E+07 |
| FNO | 7.364E+06 | 7.191E+07 |
| GALERKIN-TRANSFORMER | 5.956E+06 | 5.950E+07 |
| GNOT | 1.210E+07 | 1.631E+08 |
| LSM | 3.403E+06 | 3.040E+07 |
| FACTFORMER | 4.495E+06 | 3.968E+07 |
| DEEPLAG (OURS) | **3.052E+06** | **2.716E+07** |
| PROMOTION | 11.5% | 11.9% |

## J    Visual Result of the Boundary Condition Generalization Experiment

To demonstrate the generalizing performance of DeepLag, we visualize the showcases in Figure 26 between DeepLag and the best baseline, U-Net. This intuitive result further shows that DeepLag adaptively generalizes well on new domains and handles complex boundaries well.

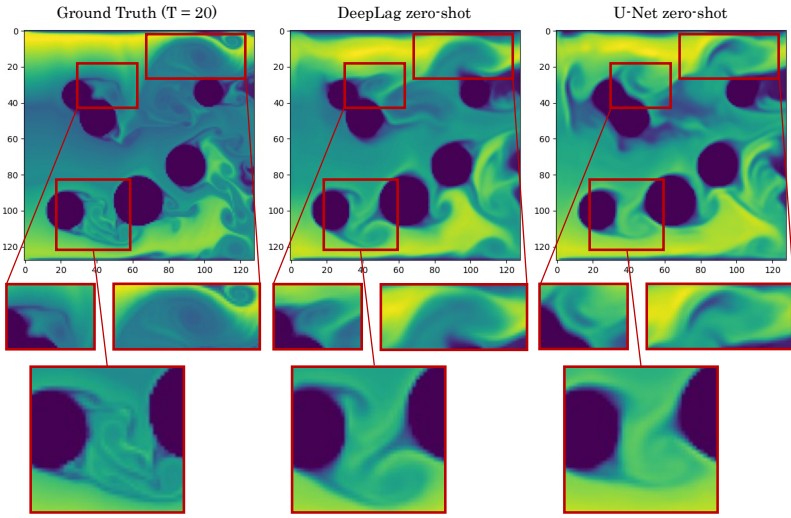

Figure 26: The visual comparison of zero-shot inference on the new Bounded Navier-Stokes.

# K    Visualization of the Movement of the Particles

We visualize the particle movements on the Bounded Navier-Stokes dataset in Figure 27. Given the dataset's complex and frequent motion patterns, we have plotted particle offsets between consecutive frames, which effectively reflect instantaneous particle movements. As depicted, DeepLag can still learn intuitive and reasonable motion without a standard physical velocity field. Notably, DeepLag performs best in this dataset, highlighting the benefits of learning Lagrangian dynamics.

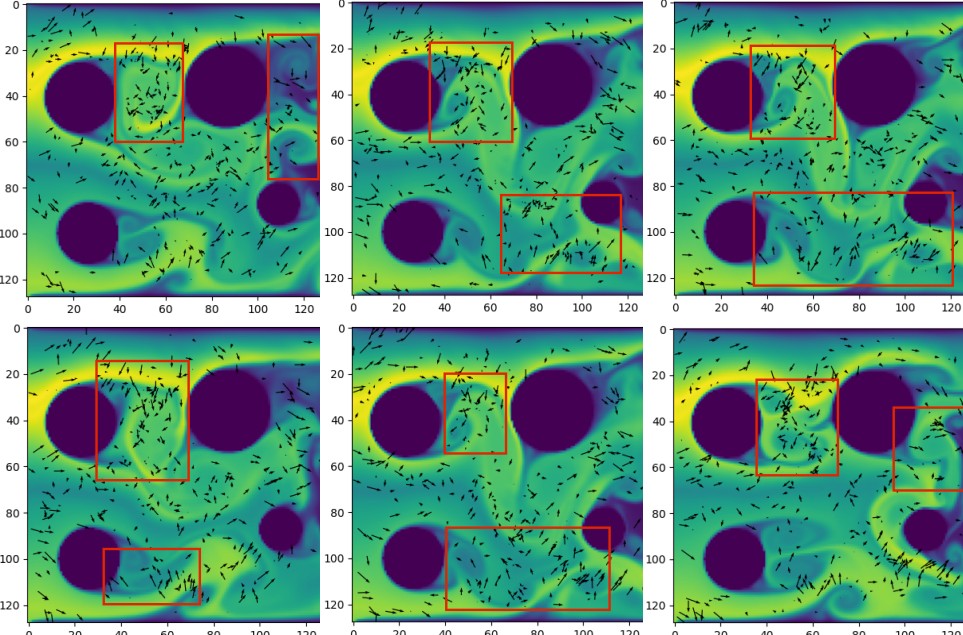

Figure 27: Visualization of the particle movements on the Bounded Navier-Stokes dataset. The plotted particle offsets between consecutive frames effectively reflect instantaneous particle movements. As depicted in the red boxes, DeepLag can accurately capture the motion mode of complex dynamics like Kármán vortex street.

