# OpenReview forum: "DeepLag: Discovering Deep Lagrangian Dynamics for Intuitive Fluid Prediction"
_NeurIPS.cc/2024/Conference — NeurIPS 2024 poster_

### Official Review · Reviewer_Kve3 · 2024-07-07

**Soundness:** 4
**Presentation:** 3
**Contribution:** 3
**Rating:** 7
**Confidence:** 4

**Summary:**

Real-world processes exhibit multi-scale spatio-temporal dynamics. Not all of this dynamics is accurately modeled by Eulerian (i.e., field-based) modeling of scientific processes and sometimes the fine-grained patterns are only able to be modeled by Lagrangian paradigms. However explicit Lagrangian-only modeling is costly and hence this paper proposes a deep learning based surrogate model that is able to jointly model the Eulerian and Lagrangian perspectives by proposing a novel `EuLag' deep learning block. The authors demonstrate the performance of the proposed architecture on multiple fluid-dynamics related tasks involving multi-phase flows.

**Strengths:**

- The authors develop a well-motivated and novel deep learning model architecture to combine the Eulerian and Lagrangian computational modeling paradigms thereby leading to improved modeling capacity owing to modeling across a larger variety of scales.

- The proposed model demonstrates performance improvements compared to state-of-the-art models like the Fourier-Neural Operator and transformer based architectures (also a central feature in the proposed EuLag block) like the Galerkin Transformer. Overall, the experimental results are convincing.

**Weaknesses:**

- More information regarding the experimental setup needs to be included in the main text of the paper. Currently, the main body of the paper (especially related to the experimental setup and the dataset descriptions, model description) cannot stand on its own without the appendix. At least the full description of the model architecture (i.e., upsampling, down sampling and other critical operations) should appear in the main text.

- Comparison with a Lagrange-only model is necessary in addition to the current field-based prediction models for a more holistic experimental comparison.

**Questions:**

1. Why have the authors not compared to a Lagrange-Only model like that of [1] for fluid flow modeling. Since the proposed paradigm is an Euler-Lagrange paradigm, baseline with models that only in the Eulerian domain seems somewhat incomplete and for completeness, comparison with one particle based deep-learning model (like [1]) should be included.


## References

[1] Sanchez-Gonzalez A, Godwin J, Pfaff T, Ying R, Leskovec J, Battaglia P. Learning to simulate complex physics with graph networks. InInternational conference on machine learning 2020 Nov 21 (pp. 8459-8468). PMLR.

**Limitations:**

Authors have addressed the potential limitations of their work sufficiently well.

---

> ### Author Rebuttal · Authors · 2024-08-07
>
> Special thanks to Reviewer Kve3 for their detailed review and insightful suggestions, your dedication to evaluating our work despite your busy schedule is greatly appreciated.
>
> > **Weakness1:** About the detailed content selected to represent in the main body of the paper.
>
> Thank you for your valuable feedback. We acknowledge that the current presentation may pose difficulties for readers seeking detailed technical insights, especially regarding the experimental setup, dataset descriptions, and model architecture. Due to space constraints, our initial focus was on presenting the core ideas and results, with specific implementation details relegated to the appendix. We will address this in our revisions by integrating a comprehensive description of the experimental setup, dataset characteristics, and detailed model architecture directly into the main body of the paper.
>
> > **Weakness2:** Comparison with a Lagrange-only model.
>
> Thank you for your comment. We appreciate your suggestion regarding including a comparison with a Lagrange-only model alongside existing field-based prediction models for a more comprehensive experimental evaluation, however, we have the following reasons not to do so:
>
> Firstly, it's important to note that existing benchmarks $^{[1,2]}$ and real-world observational datasets, such as the Ocean Current dataset by ECMWF, predominantly represent Eulerian perspectives due to their ease of collection, representation, and storage.
>
> Secondly, in practical applications like ocean studies, precisely tracking nearly infinite particles is often impractical. Therefore, there is a critical need for deep learning models that can effectively and efficiently extract Lagrangian information from Eulerian data in an unsupervised manner.
>
> Thirdly, adhering to established conventions and baseline settings allows for a fair and transparent comparison among different models. As outlined in $\underline{\text{Lines 77-78}}$ of our paper, our research is focused on Eulerian data for fluid prediction, aligning with practical applications and existing benchmarks.
>
> Regarding the comparison with a Lagrange-only model, it's important to clarify that such models operate purely on Lagrangian data inputs and outputs. This does not align with the specific problem statement and datasets used in our study, making it challenging to conduct a fair comparison.
>
> In conclusion, while we appreciate the suggestion, our experimental design focuses on evaluating field-based prediction models within the context of Eulerian data. This approach ensures consistency and relevance to practical applications and existing benchmarks.
>
> ------
>
> [1] Takamoto et al., "PDEBench: An Extensive Benchmark for Scientific Machine Learning", NeurIPS D&B 2022
> [2] Gupta and Brandstetter, "Towards Multi-spatiotemporal-scale Generalized PDE Modeling", 2022
>
> > **Q1:** Comparison with one particle-based deep-learning model
>
> Thank you for your inquiry. We appreciate your suggestion regarding comparing our approach with a Lagrange-Only model.
>
> As outlined in **Weakness2** of our paper, the decision not to include a comparison with a Lagrange-Only model stems from several considerations. Primarily, our research focuses on leveraging both Eulerian and Lagrangian perspectives within a unified framework, termed the Euler-Lagrange paradigm, currently on Eulerian data. This dual perspective allows for a more comprehensive understanding and prediction of fluid dynamics, which is the primary contribution and focus of our study.
>
> However, we acknowledge the merit of your suggestion to include comparisons with models that solely operate within the Lagrangian domain. In response to your suggestion, we have contemplated a dual approach where Eulerian information supplements purely Lagrangian particle data for prediction tasks. This idea for proving the effectiveness of the Euler-Lagrange paradigm on Lagrangian data, while relevant and insightful, falls beyond the scope of the current paper but will be considered for future work. We appreciate your valuable input and will duly note it for further exploration.

---

> > ### Comment · Reviewer_Kve3 · 2024-08-13
> > **Response to Author Rebuttal by Reviewer Kve3**
> >
> > Thank you to the authors for their responses. After consideration of the responses, I continue to remain positive about the paper and will maintain my score.

---

### Official Review · Reviewer_PRfM · 2024-07-12

**Soundness:** 3
**Presentation:** 2
**Contribution:** 3
**Rating:** 6
**Confidence:** 3

**Summary:**

The authors propose DeepLag as an approach to simulating Eulerian fluid dynamics, which makes use of Eulerian-Lagrangian co-design to improve performance. In particular, the idea is to transfer information back and forth between the Eulerian grid and initially randomly placed Lagrangian particles, which themselves exist on multiple but connected discretization scales. The intuition is that Lagrangian particles are better at capturing interactions with boundaries and, in general, tracking the location of material elements, e.g., smoke. Empirically, the gain in performance over existing models is consistent and around 10%. The limitations of the approach are its algorithmic complexity and hyperparameters.

**Strengths:**

- The idea of having "attention" between the Eulerian grid and Lagrangian particles is interesting and seems to work well.
- The multi-scale approach makes a lot of sense to map fine details to the coarser resolution. Not sure whether this is only a strength though, as this means higher algorithmic complexity of the model.
- The approach by construction offers Lagrangian particle trajectories, which might be useful in e.g. ocean dynamics research

**Weaknesses:**

- **Factual mistakes**: the authors make some very crude false statements here and there, and I'm questioning whether a person or LLM wrote sections of the manuscript, particularly the introduction.
    - Lines 24-25: "curse of dimensionality" is NOT the reason for high computational cost in fluid dynamics! Please open the Wikipedia article on the curse of dimensionality and modify the sentence in the paper.
    - Lines 40-42 state that "[Lagrangian approach] helps get around the CFL condition", which is absolutely wrong.
- **Related work taxonomy**: in section 2.2, a categorization of neural solvers is presented; however, this grouping of methods is new to me and, as much as I know, not what the community typically uses.
    - The 1st group, "ODE-based Generative Models", refers to some rather old and outdated work,
    - the 2nd group, "Neural PDE Solvers", actually only talks about Physic-Informed Neural Networks, and
    - the 3rd group, "Neural Operators for PDE", basically lists all methods that I would call modern neural PDE solvers and what is typically referred to as neural operator learning (DeepONet, FNO, etc.) is mentioned along with GNNs and CNNs.

    See PDE Bench [1] and PDE Arena [2] for a more modern categorization of common PDE learning approaches.
- **Ablations**: I'm missing an ablation on (a) the number of scales and (b) overall model size. Judging by Fig 7, in which the U-Net is 5x faster and also smaller, I wonder how a U-Net with similar (a) parameter count or (b) runtime would perform. I'm a bit worried that the claimed good performance might be just a bad hyperparameter choice of the baselines.

---
[1] Takamoto et al., "PDEBench: An Extensive Benchmark for Scientific Machine Learning", NeurIPS D&B 2022
[2] Gupta and Brandstetter, "Towards Multi-spatiotemporal-scale Generalized PDE Modeling", 2022

**Questions:**

- Fig. 7: in this figure, it looks like the U-Net has fewer parameters than DeepLag, but in lines 345-346, as well as Table 10, you state the opposite. Can you explain that?

**Limitations:**

- Thinking of Occam's Razor, it is questionable how many people would adopt the proposed approach, as it requires (a) a significantly more involved implementation effort than a typical U-Net, (b) tuning various hyperparameters like number of scales and number of particles per scale, and (c) is ~5x slower (Fig 7), while offering on average 10% performance improvement. Could the authors add some more hints on where their approach would be of practical interest? I would appreciate extending section 5 into, for example, one paragraph dedicated to the summary/strengths/applications, and one with the limitation/future work.

---

> ### Author Rebuttal · Authors · 2024-08-07
>
> Special thanks to Reviewer PRfM for their detailed review and insightful suggestions.
>
> > **strength2:** On the algorithmic complexity brought by the multi-scale design.
>
> Please recall $\underline{\text{Table 6 in Appendix A.1}}$ that the number of tracking particles at each scale has an exponential decrease w.r.t the index of that scale, resulting in adding more coarser scales only bringing about a minor fraction of extra computing, while the overall complexity doesn't vary too much. For quantitative results, please refer to **Weakness3** below.
>
> > **Weakness1:** Justification for certain statements.
>
> Thank you for carefully reviewing our paper and sorry for the controversial expressions, however, you can rest assured that every word of this paper is written by ourselves not LLM.
>
> - **Regarding 'curse of dimensionality'**, we acknowledge your concern. While the curse of dimensionality itself refers to specific challenges in high-dimensional data spaces and may not directly cause high computational costs in fluid dynamics, we understand that computational challenges in fluid dynamics primarily stem from the complex numerical methods required to solve PDEs over large and intricate grids. We will revise the text to better clarify this distinction.
>
> - **Regarding the statement of CFL condition**. We agree with you that the CFL condition is essential for ensuring numerical stability and accuracy in fluid dynamics simulations, regardless of the method employed. What we are trying to say is that Lagrangian approaches for Eulerian fluid prediction like Semi-Lagrangian method offer more flexibility in time step sizes (e.g. adaptive time stepping) than Eulerian methods. We will remove this claim for scientific rigor.
>
> > **Weakness2:** Reorganize the related works.
>
> In the previous paper, we follow the survey$^{[1]}$, where 'Neural Solvers' indicates Physic-Informed Neural Networks (PINNs) and 'Neural Operators' refers to the models that map the input function space to the output space, such as FNOs.
>
> Following your suggestion, we would rephrase category names to 'Classical ML methods', 'Physic-Informed Neural Networks (PINNs)' and 'Neural Operators'.
>
> ------
>
> [1] Physics-Informed Machine Learning: A Survey on Problems, Methods and Applications, arXiv 2022
>
> > **Weakness3:** More ablations.
>
> As per your request, we conducted further ablations, yielding the following results:
>
> | No. of scales | Relative L2 | Latent dim.   | Relative L2 |
> | - | - | - | - |
> | 1             |   0.0789    | 16            |   0.0656    |
> | 2             |   0.0658    | 32            |   0.0594    |
> | 4 (original)  | **0.0543**  | 64 (original) | **0.0543**  |
> | 5             |   0.0554    | 128           |   0.0614    |
>
> Regarding (a) the number of scales, **increasing the number of scales generally improves performance up to a point**, after which diminishing returns are observed. As for (b) overall model size, **increasing the latent dimension tends to improve performance**, but too many parameters can be unnecessary for the model, as seen in the ablations.
>
> Additionally, in response to your concern about comparing U-Net with a similar parameter count or runtime to DeepLag, please refer to $\underline{\text{Appendix E}}$. **U-Net consistently exhibits significantly larger parameter counts across all datasets compared to DeepLag**, rendering such experiments unnecessary. Furthermore, experiments scaling U-Net to match DeepLag's runtime show:
>
> | Model       | No. of Parameters | GPU Memory | Running Time | Relative L2 |
> | - | - | - | - | - |
> | U-Net-scale |    336,609,604    |   13672M   |  601s/epoch  |     NaN     |
> | DeepLag     |    19,526,827     |   12112M   |  845s/epoch  | **0.0378**  |
>
> In conclusion, augmenting U-Net with additional convolutional layers and increasing the latent dimension shows that **too many parameters can overwhelm U-Net, indicating a scalability limitation.**
>
> > **Q1:** Elaboration on Figure 7 and Table 10.
>
> Thank you for your inquiry regarding the discrepancy between $\underline{\text{Figure 7}}$ and $\underline{\text{Lines 345-346}}$, as well as $\underline{\text{Table 10}}$. Please note that $\underline{\text{Figure 7}}$ depicts **memory usage**, while $\underline{\text{Table 10}}$ and $\underline{\text{Lines 345-346}}$ refer to **parameter counts**.
>
> Thus, the confusion arises from the difference in metrics presented. Specifically, U-Net is characterized by a larger number of parameters but a smaller memory footprint, as indicated in our findings. We acknowledge the need for clarity in distinguishing these metrics and will ensure that this distinction is clearly labeled in our revised manuscript.
>
> > **Limitations:** About Occam's Razor, practical interest and writing advice.
>
> Many thanks for your valuable suggestion. Firstly, we want to highlight several properties of our model that correspond to your mentioned question:
>
> (a) About implementation effort: DeepLag can be implemented solely based on Pytorch modules, which may not be as hard as you thought.
>
> (b) About hyperparameter:  The primary consideration is the total number of tracking particles in the finest scale. Ablations in $\underline{\text{Table 5 in Section 4.4}}$ demonstrate that performance generally improves with an increased number of particles. Therefore, you can decide this up to your computation resource.
>
> (c) About efficiency: We have to say that DeepLag is slower than U-Net. However, the performance improvement brought by DeepLag can be acknowledged in some applications, such as wind tunnel test, which prefers high precision to efficiency.
>
> In addition, the learned particle trajectory can be visualized to help research understand fluid dynamics.
>
> Following your suggestion, we will extend the Conclusion to discuss more about strengths/applications and limitations/future work. One promising direction is to speed up DeepLag with advanced attention mechanisms, such as FlashAttention.

---

> > ### Comment · Reviewer_PRfM · 2024-08-12
> >
> > I thank the authors for the elaboration, new ablations, and improved content. My score accordingly increases by +1.

---

> > > ### Author Response · Authors · 2024-08-13
> > > **Thanks for Your Response and Raising the Score**
> > >
> > > Thank you for your valuable suggestions and for acknowledging our rebuttal. Your feedback has been instrumental in refining both the writing and experiments in our DeepLag paper. We greatly appreciate your recognition and support.

---

### Official Review · Reviewer_PBwE · 2024-07-15

**Soundness:** 3
**Presentation:** 3
**Contribution:** 3
**Rating:** 6
**Confidence:** 4

**Summary:**

In this paper, the author presents a Lagrangian-Eulerian hybrid paradigm to address the complexities of fluid dynamics. Instead of relying only on Eulerian observations to predict future states, we introduce DeepLag, which uncovers hidden Lagrangian dynamics within the fluid by tracking the movements of adaptively sampled key particles. In experiments, DeepLag show better performance in three demanding fluid prediction tasks both in 2D and 3D, as well as simulated and real-world fluids.

**Strengths:**

- The idea of integrating Lagrangian tracking into the deep model for assisting Eulerian based fluid prediction sounds novel.
- The LagToEu Attention and EuToLag Attention for exchanging information between Eulerian and Lagrangian view is intuitive.
- The proposed method shows better performance on all three test cases comparing to all baselines.

**Weaknesses:**

- In the Bounded Navier Stokes, the performance of 10 Frames is significant better than 30 Frames, while for the Ocean Current case, the results of  long-term rollout shows gives better result. Just wondering what is the reason between this discrepancy? Is the proposed method better than in shot-term or long-term tasks?
- How does the proposed method handle the complex boundaries? e.g., how to impose different boundary conditions into the framework?

**Questions:**

See weaknesses.

**Limitations:**

Yes. The author discuss the limitations in the paper.

---

> ### Author Rebuttal · Authors · 2024-08-07
>
> We would like to thank Reviewer PBwE for the detailed review and insightful suggestions.
>
> > **Weakness1:** Explanation of the performance difference between datasets. "Just wondering what is the reason between this discrepancy? Is the proposed method better than in short-term or long-term tasks?"
>
> Thank you for your thorough review of our paper. The observed performance differences between the Bounded Navier-Stokes and Ocean Current datasets stem from the diversity in dataset characteristics rather than any weaknesses in our model.
>
> The Bounded Navier-Stokes dataset represents a smaller scale with irregular variations, whereas the Ocean Current dataset spans larger scales with periodicity and oceanic averages. Therefore, for ocean data, achieving accurate predictions on average state can also yield excellent results over the long term.
>
> > **Weakness2:** About the boundary condition processing in the framework. "How does the proposed method handle the complex boundaries? e.g., how to impose different boundary conditions into the framework?"
>
> Actually, DeepLag is very flexible at adapting to various complex boundary conditions. As depicted in $\underline{\text{Figure 9 in Appendix G}}$, whether the boundary is known or unknown, simple or intricate, the dynamic sampling module which is optimized with the pointwise variance of vorticity could recognize the boundary area, wavefront, wake and far-field zone from the features contained in the input distribution. By the way, for datasets whose boundary is known (like the pillars in Bounded Navier-Stokes), we could concatenate the boundary mask as a new channel into model input as a guide to enhance the performance of DeepLag and the baselines.
>
> To verify the generalizing ability on the new domain of DeepLag, we ran a zero-shot test with the old model checkpoint on a newly generated Bounded N-S dataset which has a different number, position and size of obstacles. **DeepLag still has a ~7% promotion w.r.t the best baseline, U-Net, on relative L2 (DeepLag: 0.203, U-Net: 0.217). The visual comparison** ($\underline{\text{Figure 2 in Global Response PDF}}$) **between the two models further shows that DeepLag adaptively generalizes well on new domains and handles complex boundaries well**.

---

> > ### Comment · Reviewer_PBwE · 2024-08-11
> > **Response to rebuttal**
> >
> > Thanks for the authors' reply to my concerns. I don't have further questions and I remain positive about this paper.

---

> > > ### Author Response · Authors · 2024-08-11
> > > **Thanks for your response**
> > >
> > > Thank you for your prompt response—it has been very helpful to us. If there's anything further we can do to potentially improve your assessment of our paper, please don't hesitate to let us know. We would be more than happy to engage in further discussion.

---

### Official Review · Reviewer_rCcn · 2024-07-16

**Soundness:** 3
**Presentation:** 2
**Contribution:** 3
**Rating:** 5
**Confidence:** 3

**Summary:**

The authors propose a novel neural network architecture in order to leverage the advantages of the eulerian and lagrangian formalisms for fluid prediction. The so called "EuLag Block" acts on an eulerian grid based representation of the fluid as well as on a lagrangian particle based representation and enables information flow between both representations through 2 attention blocks:
1. The "LagToEu Attention" block allows the network to pass information from the lagrangian particle based representation to the eulerian grid based representation
2. The "EuToLag Attention" block allows the network to pass information from the eulerian grid based representation to the lagrangian particle based representation
On top of these 2 attention blocks, a bilinear interpolation module is used to extract information about local dynamics from the eulerian representation, which is then combined with information about global dynamics from all particles to compute the update of the particle based representation. The distribution of the initial particle positions is generated by a learnable sampling module.
The authors evaluate the resulting DeepLag architecture on 3 different fluid prediction tasks and outperform several state of the art methods. Furthermore, ablation studies were performed in order to investigate the effect of different numbers of particles, both attention blocks as well as the learnable sampling module.

**Strengths:**

- the qualitative results look convincing (especially the long term stability shown in the videos)
- fairly extensive quantitative comparison to other network architectures and ablation studies
- interesting idea to use LagToEu / EuToLag attention blocks in order to reduce the squared complexity of the attention mechanism wrt the domain size in the eulerian frame of reference to a linear complexity wrt the domain size and the number of lagrangian particles (especially if the number of particles is chosen to be relatively small).

**Weaknesses:**

- Reference to "Accelerating Eulerian Fluid Simulation With Convolutional Networks" by Tompson et al is missing. They use a particle tracer to deal with the advection term in the lagrangian frame of reference and train a CNN to perform a "pressure projection step" in the eulerian frame. They showed impressive smoke simulations in 3D similar to the experiments shown in section 4.3. Thus, I'm not sure if I can fully agree with the claim in line 55-57 that this is the first deep fluid prediction model that explicitly combines Eulerian and Lagrangian frameworks.
- Regarding the efficiency analysis (Figure 7) it would be interesting to see how a U-Net would compare to DeepLag when upscaled up to a similar running time.

**Questions:**

- How would the particle movements (shown in Figure 5 on the right) look for the Bounded Navier-Stokes dataset (Figure 4)? Would the particles still follow the velocity field although the velocity field is not considered as a field variable in this experiment but only the dye concetration?
- Could the particle movements be used to extract information about the velocity field in the Bounded Navier-Stokes dataset?
- Often in fluid dynamics (e.g. "Stable fluids" by Jos Stam or "Accelerating Eulerian Fluid Simulation With Convolutional Networks" by Tompson et al), solving the pressure field is done in the eulerian frame and requires a global solution. However, dealing with the advection term using a particle tracer requires only the local velocity field. Isn't the global attention for the particle updates a "slightly wasteful" overkill?
- How does the positional embedding look like?
- How well does your method generalize to new domains?

**Limitations:**

- How would your method scale to larger domains? I'm wondering if the fully-connected layer in the dynamic sampling module could become a bottleneck for larger domain as it seems to scale quadratically with the domain size.

---

> ### Author Rebuttal · Authors · 2024-08-07
>
> Sincerely thank Reviewer rCcn for the detailed review and insightful suggestions.
>
> > **Weakness1:** On the difference between DeepLag and FluidNet [Tompson et al, ICML 2017].
>
> Thanks for your recommendation on related work and rigorous questions. We will cite FluidNet in the revised paper. However, we have to say that FluidNet is distinct from DeepLag:
>
> |                              | FluidNet                  | DeepLag                                            |
> | - | - | - |
> | **Main Design**     | Replaces Eulerian pressure projection with CNN module | Combines Eulerian and Lagrangian in Deep Learning |
> | **Method for Lagrangian**    | **Numerical method** | Learning **deep Lagrangian info for end-to-end modeling** |
>
> The key design of FluidNet is replacing one step of classical method with a learned module implemented by a surrogate CNN for accumulation. However, in the Lagrangian perspective, it still employs the traditional numerical approach (Maccormack method).
>
> **In contrast, DeepLag represents an approach that integrates both Eulerian and Lagrangian within a pure deep learning framework**. This allows for more generalized modeling of Lagrangian dynamics across multiple elements, demonstrating its capability as a fully deep learning-based solution capable of end-to-end autonomous learning.
>
> We will revise the claim you concern to **"explicitly combines Eulerian and Lagrangian in a deep learning framework"** for scientific rigor.
>
> > **Weakness2:** Comparison between DeepLag and U-Net under similar running time.
>
> | Model       | No. of Parameters | GPU Memory | Running Time | Relative L2 |
> | - | - | - | - | - |
> | U-Net-scale |    336,609,604    |   13672M   |  601s/epoch  |     NaN     |
> | DeepLag     |    19,526,827     |   12112M   |  845s/epoch  | **0.0378**  |
>
> As per your request and following $\underline{\text{Figure 7}}$, we conducted experiments on the 3D Smoke dataset that scaled up U-Net to make a comparison between DeepLag and U-Net under similar running time. Concretely, we add more conv layers into the standard U-Net and increase the latent dimension. **The results show that too many parameters overwhelm U-Net, indicating that it has a shortcoming in scalability**.
>
>
> > **Q1:** Visualization of the particle movements of the Bounded Navier-Stokes dataset.
>
> Based on your request, we have visualized the particle movements on the Bounded Navier-Stokes dataset in $\underline{\text{Figure 1 in PDF of Global Response}}$. Given the dataset's complex and frequent motion patterns, we have plotted particle offsets between consecutive frames, which effectively reflect instantaneous particle movements.
>
> As depicted, DeepLag still can learn intuitive and reasonable motion. However, there is a slight decrease in overall quality compared to Ocean Current, which may be because of lacking standard physical quantities. However, it is worth noticing that DeepLag performs best in this dataset, which verifies the benefits of learned Lagrangian dynamics in improving prediction.
>
> > **Q2:** Velocity field extraction from particle movements of the Bounded N-S dataset.
>
> Extracting dense velocity fields from the particle perspective in the Bounded N-S dataset is challenging due to the sparse nature of particles, which is necessary to reduce computational complexity. However, **key pathways of complex motion regions can be identified through adaptive sampling, which already proves valuable for understanding fluid dynamics**.
>
> > **Q3:** Is the EuToLag attention an overkill?
>
> In theory, relying solely on local velocity fields for particle updates seems sufficient. However, in practical fluid dynamics, the exact velocity of fluid motion is often unknown, and due to the incompressibility of fluids, disturbances from distant regions can influence local dynamics. Therefore, incorporating global attention mechanisms ensures comprehensive modeling, addressing scenarios where distant influences play a role. Take Ocean Current as an instance, the test set Relative L2 increases to 0.0264 from 0.0250 after removing the EuToLag attention (at the 20th training epoch).
>
> Moreover, the EuToLag attention introduces minimal computational overhead ($O(n)$) and slightly increases training time (from 1030s/epoch to 1150s/epoch on Bounded N-S), while yielding noticeable performance improvements. Thus, we believe that EuToLag attention is not 'overkill'.
>
> > **Q4:** About the positional embedding.
>
> We concatenate two additional channels to the input (three for 3D Smoke), representing normalized (x, y) coordinates (or (x, y, z) for 3D Smoke).
>
> > **Q5:** Generalization on new domains.
>
> To verify the generalizing ability of DeepLag, we ran a zero-shot test with the old model checkpoint on a newly generated Bounded N-S dataset which has a different number, position and size of obstacles.
>
> DeepLag still has a ~7% promotion w.r.t the best baseline, U-Net, on relative L2 (DeepLag: 0.203, U-Net: 0.217). The visual comparison ($\underline{\text{Figure 2 in Global Response PDF}}$) between the two models further shows that DeepLag adaptively generalizes well on new domains.
>
> > **Limitations:** On the scalability to larger domains of the fully-connected layer in the dynamic sampling module.
>
> Sorry for the vague description. The "fully-connected layer" in $\underline{\text{Line 151}}$ **refers to channel dimension MLP rather than spatial.** Therefore, the quadratical complexity for a fully-connected layer doesn't exist.
>
> Further, we trained a new model on a 256*256 Bounded N-S (4x larger domain) as requested, which still has a ~17% promotion w.r.t U-Net on relative L2 (DeepLag: 0.051, U-Net: 0.060), the comparison below also shows that the increase of the time complexity is minor, underscoring the scalability of DeepLag.
>
> | Resolution | GPU Memory | Running Time | Relative L2 |
> | - | - | - | - |
> | 128*128    |   5420M    | ~1150s/epoch |   0.0543    |
> | 256*256    |   13916M   | ~1300s/epoch |   0.0514    |

---

### Official Review · Reviewer_UohQ · 2024-07-18

**Soundness:** 3
**Presentation:** 2
**Contribution:** 3
**Rating:** 6
**Confidence:** 3

**Summary:**

The paper introduces a novel approach to predicting fluid dynamics by integrating both Lagrangian and Eulerian paradigms. The model, named ‘DeepLag,’ utilizes transformer blocks to process and integrate information from Eulerian and Lagrangian perspectives. Initially, DeepLag predicts the future state of the Eulerian space. Subsequently, it uses this prediction to infer the Lagrangian movements of key tracked particles. This methodology allows the Lagrangian dynamics to inform and guide the evolution of the Eulerian predictions, enhancing the overall prediction accuracy.

DeepLag is evaluated against various baselines across three diverse datasets, including simulated bounded Navier-Stokes equations, real-world ocean currents, and 3D smoke dynamics. These experiments demonstrate the model’s effectiveness in handling both 2D and 3D fluid dynamics in simulated and real-world scenarios.

**Strengths:**

- The DeepLag model introduces a novel framework that effectively combines Eulerian and Lagrangian perspectives, allowing for dynamic propagation in both spaces. This dual approach is innovative in fluid dynamics modeling, enhancing the prediction accuracy by leveraging the strengths of both paradigms.
- The model’s effectiveness is rigorously tested across three challenging datasets—bounded Navier-Stokes equations, real-world ocean currents, and 3D smoke dynamics. The experiments demonstrate superior performance not only in standard scenarios but also in both short-term and long-term prediction tasks, highlighting the model’s versatility and robustness.
- One of the standout features of DeepLag is its ability to provide interpretable results by showcasing individual particle trajectories via Lagrangian dynamics. This aspect is particularly valuable as it not only enhances the understanding of fluid movements but also aids in validating the model’s predictions through visual and traceable particle paths.

**Weaknesses:**

- The model’s reliance on extensive hyperparameter tuning for particle tracking could pose challenges in terms of replicability and efficiency. This complexity might limit the accessibility of the model for practical applications without substantial computational resources.
- Additionally, the particle tracking mechanism struggles to maintain focus on particles near the domain borders. This limitation is particularly concerning in scenarios where significant dynamic changes occur near these borders, such as the presence of obstacles. The inability to track these dynamics could potentially lead to incomplete or inaccurate modeling of fluid behavior in such areas.
- The paper contains a few grammatical errors that, while minor, could detract from its overall professional presentation.

**Questions:**

- How to interpret the difference between the left and right parts of Fig. 1?
- Is there supervision over the Lagrangian space, or does it only happen in the Euler space?
- Would interchanging the positions of the LagToEu and EuToLag blocks affect the model’s performance or learning dynamics? If so, how?
- Given that Table 5 shows only a minimal performance boost from the EuToLag attention, how should this be interpreted in the context of the model’s overall efficiency and effectiveness?
- In the Bounded Navier-Stokes experiments, are the positions of the cylinders consistent across all trials within the dataset, or do they vary?

**Limitations:**

Limitations have been addressed

---

> ### Author Rebuttal · Authors · 2024-08-07
>
> Many thanks to Reviewer UohQ for the detailed review and suggestions.
>
> > **Weakness1:** The model’s reliance on extensive hyperparameter tuning for particle tracking could pose challenges.
>
> **The only hyperparameter needed for tuning is the total number of the tracking particles in the finest scale.** Please recall $\underline{\text{Table 6 in Appendix A.1}}$ that the number of tracking particles at each scale has an exponential decrease w.r.t the index of that scale. The ablation in $\underline{\text{Table 5}}$ shows that the performance always tends to incline as we simply increase this number (which means our DeepLag is scalable), therefore **the hyperparameter choice depends more on the limitation of computing resources rather than blind searching**.
>
> > **Weakness2:** The particle tracking mechanism struggles to maintain focus on particles near the domain borders.
>
> We respectfully point out that our model works well on boundary particles. $\underline{\text{Figure 9 in Appendix G}}$ shows that **DeepLag is apt to sample more points in the complex-dynamics area**, which is near and behind the column obstacles. Plus, the red boxes in $\underline{\text{Figure 10,14 in Appendix H and Figure 16 in Appendix I}}$ indicate that **DeepLag is good at capturing the fine details in the wake zone where turbulence and vortex form**.
>
> > **Weakness3:** The paper contains a few grammatical errors.
>
> Thank you for your thorough review of our paper and for bringing to our attention the grammatical errors that may detract from its professional presentation. We sincerely apologize for these inaccuracies and have already taken steps to correct them. Specifically, we will address errors such as the **article misusage** in $\underline{\text{Line 3 of paper}}$ ("an Eulerian perspective" corrected to "the Eulerian perspective") and the **preposition error** in $\underline{\text{Line 70 of paper}}$ ("tracking a certain particle with initial position $\mathbf{s}_0$ with its displacement $\mathbf{d} = \mathbf{d}(\mathbf{s}_0, \mathit{t})$" corrected to "tracking a particular particle from initial position $\mathbf{s}_0$ by its displacement $\mathbf{d} = \mathbf{d}(\mathbf{s}_0, \mathit{t})$"). These corrections are crucial for maintaining the professional standard of our presentation. Moving forward, we will conduct a comprehensive proofreading to ensure that all grammatical errors are identified and rectified.
>
> > **Q1:** About the difference between the two parts of Fig. 1
>
> As indicated in the figure caption, Fig. 1 presents two distinct visualizations. The left part depicts the trajectories of Lagrangian particles overlaid on the mean state observed from the Eulerian perspective. In contrast, the right part displays successive Eulerian frames with scattered positions of tracked particles. These visualizations align with the description provided in $\underline{\text{Lines 42-43 of paper}}$, where **the trajectories of fluid motion are more visibly represented through the dynamic Lagrangian view compared to the density variations observed in static Eulerian grids**. This underscores our motivation to incorporate and study Lagrangian dynamics.
>
> > **Q2:** About the supervision over the Lagrangian space.
>
> There is no supervision signal in the Lagrangian space due to the following reasons. Firstly, the widely used large-scale and fine-grained data in hydromechanics is all Eulerian since they're relatively easier to collect, represent and store. Secondly, it's impractical to track almost infinite particles precisely in real-world applications like ocean study, so **there is a need for deep models to unsupervised extract Lagrangian information from Eulerian data effectively and efficiently**. Thirdly, following the convention and the setting of baselines allow us a fair and clear comparison.
>
> > **Q3:** The result of interchanging the positions of the LagToEu and EuToLag blocks.
>
> | Dataset / Model | Original | Swap EuToLag and LagToEu |
> | - | - | - |
> | Bounded N-S     |  0.0543  |          0.0545          |
> | 3D Smoke        |  0.0378  |          0.0378          |
>
> We conducted experiments as per your request, swapping the positions of the EuToLag and LagToEu blocks and validating their effects on both 2D (Bounded Navier-Stokes) and 3D (3D Smoke) datasets. The experimental results indicate that **there was minimal change in performance, suggesting that the flow of information between Eulerian and Lagrangian perspectives is bidirectional**. This insensitivity to the order in which information is transferred between the two perspectives underscores the robustness of our approach. These findings further support our statement in $\underline{\text{Line 73}}$ of the manuscript that "**Two perspectives are constitutionally equivalent**," which is substantiated both theoretically ($\underline{\text{Eq. (1) and (2) in Section 2.1}}$​) and intuitively.
>
> > **Q4:** Explanation on the ablation of the EuToLag attention.
>
> The relatively small promotion of EuToLag attention can be interpreted as follows.
>
> Firstly, **EuToLag attention itself adds minimal computational overhead ($O(n)$) and only marginally increases the training time (from 1030s/epoch to 1150s/epoch)**. The primary computational load lies in the LagToEu module, which processes dense Eulerian data using patch embedding and patch recovery. Therefore, EuToLag attention does not significantly impact efficiency.
>
> Secondly, **in scenarios where precision demands capturing global information for effectiveness, integrating EuToLag attention remains justified**. In this paper, we mainly focus on effective Eulerian-Lagrangian collaborative modeling and efficiency optimization is a promising future work.
>
> > **Q5:** Positions of the cylinders in the Bounded Navier-Stokes Benchmark.
>
> The positions of the cylinders are fixed, but the initial condition varies in different samples, which can simulate a scenario like bridge pillars in a torrential river.

---

> > ### Comment · Reviewer_UohQ · 2024-08-13
> >
> > Thank you to the authors for providing additional experiments and clarifications in response to my comments. I appreciate these efforts and will maintain my score.

---

### Author Rebuttal · Authors · 2024-08-07

## Global Response and Summary of Revisions

We sincerely thank all the reviewers for their insightful reviews and valuable comments, which are instructive for us to improve our paper further.

This paper proposes a **new deep Eulerian-Lagrangian hybrid paradigm for fluid prediction**, DeepLag, which can benefit from the advantages of both perspectives. DeepLag is derived from the equivalent and complementary theory of the two perspectives as a practically novel end-to-end deep learning framework, which **consistently boosts the performance of three difficult benchmarks, covering both 2D and 3D fluid.**. Detailed visualizations and model analysis are provided.

The reviewers generally held positive opinions of our paper, in that the proposed method is "**innovative**", "**effectively combines Eulerian and Lagrangian perspectives**", "**particularly valuable**", and "**versatile and robust**"; The model’s effectiveness is "**rigorously tested**" and demonstrated "**superior performance not only in standard scenarios but also in both short-term and long-term prediction tasks**".

The reviewers also raised insightful concerns and constructive questions. We made every effort to address all of them by providing detailed clarification, requested results and visualizations. Here is the summary of the major revisions:

- **Explain hyperparameter choosing for particle tracking would not pose challenges (Reviewer UohQ, PRfM):** Firstly, we point out that the only hyperparameter needed for tuning is the total number of the tracking particles in the finest scale. Secondly, we recall the hyperparameter details in the appendix and the ablation result to explain that the hyperparameter choice depends more on the limitation of computing resources rather than blind searching.
- **Clarify the difference between DeepLag and FluidNet [ICML 2017] (Reviewer rCcn):** Firstly, we emphasize that FluidNet is mainly based on the numerical method, which replaces Eulerian pressure projection with a CNN module, while DeepLag integrates both Eulerian and Lagrangian perspectives within a pure deep learning framework, which learns deep Eulerian and Lagrangian info for end-to-end modeling. Secondly, we will revise the claim to "explicitly combines Eulerian and Lagrangian in a deep learning framework" for scientific rigor.
- **Experiments of scaling up U-Net and comparing it with Deepag under similar running time (Reviewer rCcn, PRfM):** Following the reviewers' request and the efficiency analysis in our paper, we conducted experiments on the 3D Smoke dataset that scaling up U-Net to make a comparison between DeepLag and U-Net under similar running time. The results show that too many parameters overwhelm U-Net, indicating that it has a shortcoming in scalability.
- **Experiments of generalization on new domains and boundary conditions (Reviewer rCcn, PBwE):** Following the reviewers' concern, we ran a zero-shot test with the model checkpoint trained on the original dataset, on a newly generated Bounded N-S dataset which has a different number, position and size of obstacles, to verify the generalizing ability of DeepLag. Both quantitive and visual result shows that DeepLag adaptively generalizes well on new domains and complex boundary conditions.
- **Correct writing issues (Reviewer UohQ, Kve3):** We sincerely thank the reviewers for the valuable feedback from the careful reviewers and explain that the limitations on space constraints make us move some detailed content to the appendix. And we promise to conduct a comprehensive proofreading to ensure that all grammatical errors are resolved and include a comprehensive description of the experimental setup, dataset, and model architecture in the main text.

The valuable suggestions from reviewers are very helpful for us to revise the paper to a better shape. All the above revisions will be included in the final paper. We'd be very happy to answer any further questions.

Looking forward to the reviewer's feedback.

#### **The mentioned materials are included in the following PDE file.**

- **Figure 1 (Reviewer rCcn)**: Visualization of the particle movements on the Bounded Navier-Stokes dataset.
- **Figure 2 (Reviewer rCcn, PBwE)**: The visual comparison of zero-shot inference on the new Bounded Navier-Stokes dataset between the best baseline, U-Net, and DeepLag.

---

### Decision · Program_Chairs · 2024-09-25

**Decision:**

Accept (poster)

**Comment:**

The submitted paper proposes a method for the prediction of fluid dynamics by particle tracking and integration of Eulerian and Lagrangian perspectives. The five experts appreciated novelty, evaluation, performance, and interpretability.

Some minor weaknesses were identified (hyper parameter tuning, dependency on tracking, behavior close to boundaries, some  writing issues and technical problems in formulations), but which do not severely impact readiness for publication, in particular given the authors' answers, which made some reviewers increase their scores.

The AC concurs.